# $DPOT_{L_0}$: Concealing Backdoored model updates in Federated Learning by Data Poisoning with $L_0$-norm-bounded Optimized Triggers

## Abstract

Traditional backdoor attacks in Federated Learning (FL) that rely on fixed trigger patterns and model poisoning exhibit deficient attacking performance against state-of-the-art defenses due to the significant divergence between malicious and benign client model updates. To effectively conceal malicious model updates among benign ones, we propose $DPOT_{L_0}$, a backdoor attack strategy in FL that dynamically constructs a per-round backdoor objective by optimizing an $L_0$-norm-bounded backdoor trigger, making backdoor data have minimal effect on model updates and preserving the global model's main-task performance. We theoretically justify the concealment property of $DPOT_{L_0}$'s model updates in linear models. Our experiments show that $DPOT_{L_0}$, via only a *data*-poisoning attack, effectively undermines state-of-the-art defenses and outperforms existing backdoor attack techniques on various datasets.

## 1. Introduction

Federated Learning (FL) is a decentralized machine-learning approach that has gained widespread attention recently. Unlike traditional centralized model training, FL synthesizes model updates contributed by multiple clients, each computed locally from that client's data. That is, in each round of FL, a central server distributes a global model to participating clients, each of whom independently trains the model on its local data, and its model updates are aggregated by the server to update the global model. This approach offers enhanced data privacy, reduced communication overhead, and scalability for a large number of clients. Despite its advantages, FL has been proven susceptible to backdoor attacks (Bagdasaryan et al., 2020). In FL, backdoor attacks occur when adversaries inject triggers into a subset of clients' local data, causing their local models trained on the poisoned data to become compromised. After aggregating these compromised local models, the global model produces adversary-desired results when the same trigger conditions are met. In this work, we term clients manipulated by adversaries during local training as *malicious clients*, and those unaffected as *benign clients*.

Traditional backdoor attacks in FL present two common deficiencies. First, the patterns of backdoor triggers are pre-defined by the attacker and remain unchanged throughout the entire attack process (Bagdasaryan et al., 2020). Consequently, the learning objective brought by backdoored data is static and incoherent with the learning objective of main-task data (benign objective), resulting in distinct differences in model updates after training. These malicious clients' model updates are therefore easily canceled out by robust aggregations. Second, many approaches rely on model-poisoning techniques to enhance the effectiveness of backdoor attacks. Implementing model-poisoning attacks requires attackers to change the training procedures of a certain number of clients to make their local training algorithms different from other clients. However, achieving this condition is challenging, as advanced defense mechanisms (Riege et al., 2024) have introduced Trusted Execution Environments (TEEs) to ensure the secure execution of client-side training, making it harder to maliciously modify the training procedure.

Existing defenses against backdoor attacks in FL rely on a hypothesis that backdoor attacks will cause the updating direction of a model to deviate from its original benign objective (Fung et al., 2020; Cao et al., 2021). To counter this hypothesis, adversaries can align models' malicious updating directions to their original benign objectives. Applying this idea to FL, if the injection of backdoored data has minimal effect on a client's model updates, then detecting this client as malicious becomes challenging for defenses based on analyzing clients' model updates.

Inspired by testing-stage adversarial exsamples (Szegedy, 2014; Carlini & Wagner, 2017), recent studies on backdoor attacks in FL have proposed adding adversarial perturbations to client data to minimize their impact on model updates, which we term it as $L_2$-norm-bounded optimized triggers (Nguyen et al., 2024; Lyu et al., 2023). However, adding perturbations to data does not produce consistent backdoor features for the model to learn; instead, it substantially alters benign features, transforming them into new features to associate with target labels. Our experimental results indicate that this will overdetermine the learning objec-

tive, hindering the long-term convergence of the main-task objective. Our comparison experiments also demonstrate that pixel-pattern triggers with consistent backdoor features are generally more effective for data-poisoning-only attacks in FL.

In this work, we propose **D**ata **P**oisoning with $L_0$-norm-bounded **O**ptimized **T**rigger ($DPOT_{L_0}$), a backdoor attack on FL that dynamically constructs the malicious objective to align model updates to the benign objective. $DPOT_{L_0}$ optimizes a single $L_0$-norm-bounded backdoor trigger with consistent appearance across different images, aiming to introduce the fewest possible features to the learning objective. We provide theoretical justification that the difference in model update directions for benign and malicious objectives can be minimized by reducing the error of the malicious data on the model. Our experiments demonstrate that these small differences brought by $DPOT_{L_0}$ enable malicious model updates to bypass defenses and integrate into global models, resulting in backdoored global models. Compared to existing optimized triggers, $DPOT_{L_0}$ empirically proves to be a more effective training-stage attack, demonstrating better attack effectiveness and main-task objective convergence. We ensured $DPOT_{L_0}$'s subtlety by constraining the number of trigger pixels, degrading the accuracy of a clean vision model when classifying the poisoned data by no more than 30%.

Unlike testing-stage $L_0$-norm-bounded adversarial examples (Papernot et al., 2016), the $DPOT_{L_0}$ trigger is required to maintain a consistent appearance across different images, serving as the unified backdoor feature. In this work, we proposed algorithms to optimize the pixel values and placements of a specified-size trigger using a set of client data and a global model as input. To the best of our knowledge, this is the first work to address this challenge. Without any assistance of model-poisoning techniques, $DPOT_{L_0}$ is an attack conducted simply by executing a normal training process on the poisoned data containing the $DPOT_{L_0}$ trigger.

We evaluated $DPOT_{L_0}$ on four image datasets (FashionMNIST, FEMNIST, CIFAR10, and Tiny ImageNet) and four model architectures including ResNet and VGGNet. We assessed the attack effectiveness of $DPOT_{L_0}$ under a variety of defense conditions, testing it against twelve defense strategies that are based on analyzing clients' model updates along with one defense strategy that uses client-side adversarial training to recover the global model (Zhang et al., 2023). We compared $DPOT_{L_0}$ attack with four state-of-the-art data-poisoning backdoor attacks that employ fixed-pattern triggers, distributed fixed-pattern triggers (Xie et al., 2020), partially $L_0$-norm-bounded optimized triggers (Zhang et al., 2024), and $L_2$-norm-bounded optimzed triggers (Nguyen et al., 2024). Using a small number of malicious clients (5% of the total), $DPOT_{L_0}$ outperformed existing data-

poisoning backdoor attacks in effectively undermining defenses without affecting the main-task performance of the FL system.

## 2. Related Work

### 2.1. Backdoor Attacks in FL

FL is very vulnerable to backdoor attacks. As training data are privately held by clients, the security of data is hard to track and protect. We discussed existing backdoor attacks in FL for image classification tasks based on some important properties (more details can be found in Appendix A.2).

**With vs. Without model poisoning.** Backdoor attacks in FL primarily rely on data poisoning, where attackers embed triggers in local training data and alter labels to train malicious models. Model poisoning (Fang et al., 2020) is often introduced to strengthen these attacks, by directly manipulating clients' model updates or training algorithms. However, Trusted Execution Environments (TEEs), which authenticate and protect client-side training, make model poisoning difficult. In contrast, data poisoning is easier for attackers to conduct and harder to prevent, as clients would gather data from open, vulnerable sources.

**Static objective vs. Dynamic objective.** A static objective in backdoor attack represents a pre-defined and unchanging objective that is independent of the training system's status, such as associating certain input features or patterns with incorrect predictions. Having static objectives make malicious model updates easier to detect due to their inconsistency with main-task objective. In contrast, a backdoor attack that adjusts its objective based on the training system's status is referred to as having a dynamic objective. For example, Gong et al. (2022) and Fang & Chen (2023) optimized the trigger pattern based on a hypothesis that maximizing the activation of certain neurons in the backdoored local model can enhance the attack's persistence on the global model, which provides preliminary insights into the potential of dynamically changing backdoor objectives. Zhang et al. (2024) optimized triggers for a situation where the global model is directly trained to unlearn the trigger, which is another pioneering work exploring the potential of using dynamic objectives to attack FL.

$L_2$**-norm vs. $L_0$-norm bounded optimized trigger.** Existing works (Lyu et al., 2023; Nguyen et al., 2024) proposed to conceal malicious model updates by using adversarial examples (Goodfellow et al., 2015; Kurakin et al., 2017) as poisoned data, and the perturbations on these examples are referred to as $L_2$-norm-bounded optimized triggers. However, while adversarial examples are effective as testing-stage attack techniques, they are less suited for training-stage backdoor attacks. The extensive inconsistency introduced by adversarial alterations creates numerous redundant fea-

tures, overdetermining the learning objective and hindering the convergence of the benign objective. Recent studies on $L_0$-norm-bounded optimized triggers (Zhang et al., 2024; Fang & Chen, 2023) made constructive attempts in optimizing the values of fixed-shape triggers alongside their attack strategies. $DPOT_{L_0}$ enhances the effectiveness of the $L_0$-norm-bounded optimized trigger by optimizing not only its values but also its shape and placement. This enhancement enables data-poisoning attacks to achieve better performance without relying on additional attack strategies.

**2.2. Defenses against Backdoor Attacks in FL**

In this work, we focus on defenses that adhere to the privacy-preserving principles of FL originally introduced by McMahan et al. (2017): clients' private data are kept local, and their model updates are not shared with any entities other than the server. For a discussion on additional defenses with varying privacy-preserving properties, please refer to the Appendix A.3.

In existing defenses, the server and clients are the two subjects commonly considered for implementing defense strategies. For benign clients as the defense subject, the global model of each round is the input they receive from the FL system. Inspired by Neural Cleanse (Wang et al., 2019), Zhang et al. (2023) proposed using trigger inversion on the global model and adversarial training on local models to mitigate the impact of the backdoor trigger. However, its effectiveness against continually evolving optimized triggers remains unaddressed. For server as the defense subject, clients' model updates are the input that the server receives from the FL system. Numerous studies proposed to defend against backdoor attacks by analyzing clients' model updates, which can be further classified into the two categories below.

**Excluding model updates with outlier values or characteristics.** Some existing works presume that a malicious client's model updates will exhibit significant differences from those of benign clients in values or certain characteristics extracted from values. Nguyen et al. (2022) and Fung et al. (2020) exclude a client's model updates that have outlier cosine similarity to other clients' model updates. Sharma et al. (2023) and Ozdayi et al. (2021) reduce or penalize the contribution of model updates that show a certain degree of sign dissimilarity, either on a model-wise or parameter-wise basis. Kumari et al. (2023) and Fereidooni et al. (2024) assess the probabilistic distribution and frequency transformation of clients' model updates, and eliminate outliers in these characteristics. Mozaffari et al. (2023) create a sparse space of model updates for clients to vote, and the server rejects outlier votes and aggregates the rest.

**Byzantine-robust aggregation.** Some existing works pro-

pose aggregating only the most trustworthy model updates to tolerate the presence of malicious clients. Yin et al. (2018) aggregate reliable model updates parameter-wise by taking median or trimmed mean, while Blanchard et al. (2017), (Cao et al., 2022), and Pillutla et al. (2022) select and aggregate reliable model updates model-wise.

Analyzing clients' model updates can effectively defend against backdoor attacks that cause distinctions between malicious clients' and benign clients' model updates. However, when a backdoor attack can conceal malicious clients' model updates among benign ones, defenses based on this strategy will struggle (Bagdasaryan et al., 2020). In this work, we show that this goal can be achieved by dynamically changing the backdoor objectives defined on poison data.

# 3. Threat Model

**Attacker's capability and background knowledge:** As shown in Figure 1, we assume that each FL client—even a malicious one—is equipped with trustworthy training software that conducts correct model training on the client's local training data and transmits the model updates to the FL server. Aligning with the security settings in the state-of-the-art defense work (Riege et al., 2024), we assume that both the client training pipeline and the FL server, as well as the communication between them, faithfully serve FL's main task training and cannot be undetectably manipulated. These properties would be achievable by executing FL training within Trusted Execution Environments (TEEs) (Schneider et al., 2022; Riege et al., 2024), for example, by applying cryptographic protections to the updates (e.g., a digital signatures) to enable the FL server to authenticate the updates as coming from the TEEs.

The capability of malicious clients in our attack is limited to the manipulation of their local training data that are input to their training pipelines. In addition, in line with existing works (Lyu et al., 2023; Zhang et al., 2024; Fang & Chen, 2023; Gong et al., 2022), we do not assume the secrecy of the global model provided by the FL server, as it would typically need to be accessible outside TEEs for use in local inference tasks. As such, in each FL round, clients are granted white-box access to the global model. Originating from initially benign clients that have been compromised, these malicious clients possess some local training data for the FL main task as background knowledge.

**Attacker's goals:** The malicious clients aim to accomplish the following goals.

- **Effectiveness**. For classification tasks, *Attack Success Rate* ($ASR$) is the accuracy of a model in classifying data with triggers into a target label. In FL, backdoor attacks aim to make the post-aggregation global model misclassify data with training-stage triggers into a target

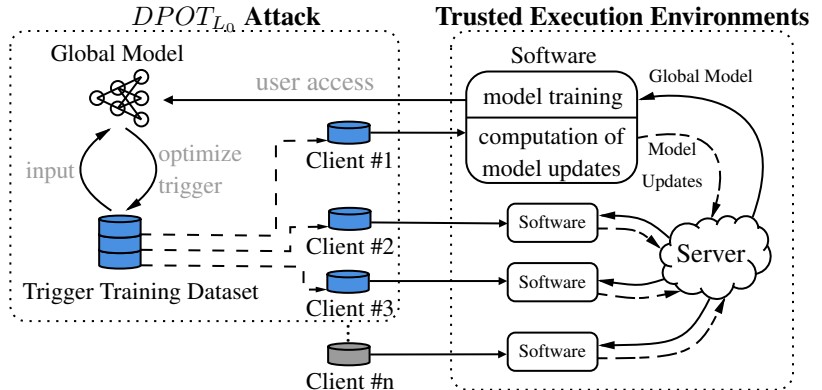

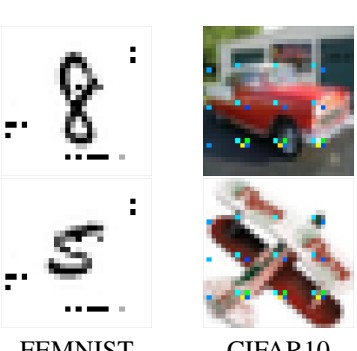

FEMNIST        CIFAR10

*Figure 1.* Overview of $DPOT_{L_0}$ attack process on a FL system within Trusted Execution Environments (TEEs). In this figure, Client #1, #2, and #3 perform as the malicious clients while other clients (e.g. Client #n) are benign clients.

*Figure 2.* Data with $DPOT_{L_0}$ triggers.

label. We evaluate a backdoor attack's final effectiveness using the final-round global model's $ASR$, with 50% as the success threshold. By-round effectiveness is measured as the average $ASR$ across all FL rounds. Combined with the final $ASR$, it indicates how quickly an attack achieves sufficient effectiveness.

- **Main-task Convergence**. The main-task convergence goal of a backdoor attack is to preserve the global model's accuracy on its main-task data at a normal level, ensuring the model's functionality remains intact and the attack goes unnoticed.
- **Subtlety**. Backdoor triggers should preserve data's main details and avoid causing misinterpretation (Figure 2). Subtlety can be evaluated by measuring accuracy drops using an un-attacked computer vision model. We aim to ensure that accuracy drops by no more than 30%.

## 4. $DPOT_{L_0}$ **Design**

### 4.1. Building a Trigger Training Dataset

At the beginning of the $DPOT_{L_0}$ attack, we initially gather all available benign data from the malicious clients' local training datasets and assign a pre-defined target label $y_t$ to them. We refer to this new dataset, which associates benign data with the target label, as the trigger training dataset $D$.

### 4.2. Determining Trigger Size ($tri_{size}$)

We determine the trigger size by ensuring that the accuracy drop of the poisoned data predicted as benign by an un-attacked model does not exceed a threshold, which we set at 30% in this work. Different model architectures and datasets result in varying trigger sizes under this standard. Adversaries determine $tri_{size}$ based on their background knowledge of the FL model and data, and can dynamically reduce it during FL training to enhance the trigger's subtlety, with a trade-off in $ASR$. In the following discussion, we

assume $tri_{size}$ is a static and optimal value.

### 4.3. Optimizing a Backdoor Trigger

We independently generate a backdoor trigger for each round's data poisoning using the optimization algorithms 1 and 2.

In the image classification context, consider the global model $W_g$ as the input and all pixels within an image forming the parameter space. Our approach seeks to identify a subset of parameters that have the greatest impact on producing the malicious output (i.e., the target label), and then optimizes the values of these parameters to further improve the accuracy of the result. The pixels in this subset with their optimized values will serve as a backdoor trigger. To enhance generalization performance of this trigger, we use all images in the trigger training dataset $D$ to optimize its pixel placements and pixel values.

---

**Algorithm 1** Computation for Trigger Location

**Input:** $W_g$, $D$, $y_t$, $tri_{size}$
**Output:** $E_t$
1: $\forall x \in D : y_x \leftarrow W_g(x)$.
2: $\mathcal{L} \leftarrow \frac{1}{|D|} \sum_{x \in D} (y_x - y_t)^2$.
3: $\forall x \in D : \delta_x \leftarrow \frac{\partial \mathcal{L}}{\partial x}$.
4: $\delta \leftarrow abs(\sum_{x \in D} \delta_x)$.
5: $\delta_f \leftarrow$ flatten $\delta$ into a one-dimensional array.
6: $S \leftarrow argsort(\delta_f)$. {Store the sorted indices (descending sort)}
7: $E_t \leftarrow S[: tri_{size}]$. {Top $tri_{size}$ indices are trigger locations}
8: $E_t \leftarrow$ transform from one-dimensional indices to indices for $x \in D$.

---

**Compute trigger-pixel placements** $E_t$**.** In Algorithm 1, we select pixel locations that contain the largest absolute gradient sum with respect to the backdoor objective as the trigger-pixel placements. Algorithm 1 takes inputs including the global model $W_g$, the trigger training dataset $D$, the target label $y_t$, and a parameter $tri_{size}$ that specifies the

trigger size. The trigger size $tri_{size}$ determines the number of pixel locations we will choose. The output of the Algorithm 1 is the trigger-pixel placement information denoted as $E_t$.

We calculate the loss of the global model $W_g$ on clean images in dataset $D$ predicted as the target label $y_t$, using Mean Square Error (MSE) as the example loss function. Gradients of the loss with respect to each pixel are computed and summed across all images, producing an absolute gradient matrix. This matrix is flattened, sorted in descending order, and the top $tri_{size}$ indices are identified as the trigger-pixel placements, which are then mapped back to the original image shape.

---

**Algorithm 2** Optimization for Trigger Value

---

**Input:** $E_t, W_g, D, y_t, n_{iter}, \gamma$
**Output:** $V_t$
1: **for** $iteration \leftarrow 1$ to $n_{iter}$ **do**
2:  $D' \leftarrow D$.
3:  **if** $iteration = 1$ **then**
4:    $V_t \leftarrow \frac{1}{|D'|} \sum_{x \in D'} x$.
5:  **else if** $iteration > 1$ **then**
6:    $\forall x \in D' : x[E_t] \leftarrow V_t[E_t]$.
7:  **end if**
8:  $\forall x \in D' : y_x \leftarrow W_g(x)$.
9:  $\mathcal{L} \leftarrow \frac{1}{|D'|} \sum_{x \in D'} (y_x - y_t)^2$.
10:  $\forall x \in D' : \delta_x \leftarrow \frac{\partial \mathcal{L}}{\partial x}$.
11:  $\delta \leftarrow \sum_{x \in D'} \delta_x$.
12:  $V_t[E_t] \leftarrow (V_t - \gamma \cdot \delta)[E_t]$.
13: **end for**

---

**Optimize trigger-pixel values $V_t$.** In Algorithm 2, we optimize the values of the trigger pixels defined in $E_t$ using a learning-based approach. Algorithm 2 requires the following inputs: the trigger-pixel placements $E_t$, the global model $W_g$, the trigger training dataset $D$, and the target label $y_t$. Additionally, it uses two training parameters: the number of training iterations $n_{iter}$ and the learning rate $\gamma$. The output produced by Algorithm 2 is the trigger-pixel value information denoted as $V_t$.

In each iteration, we create a copy dataset $D'$ of the clean dataset $D$ to embed the optimized trigger. In the first iteration, we initialize the trigger-pixel value matrix $V_t$ by averaging pixel values across all images in $D'$. We then compute the loss of the global model $W_g$ on images from $D'$ with the target label $y_t$, followed by calculating the gradients of the loss with respect to each pixel, storing them in $\delta_x$. The gradient matrix $\delta$ is obtained by summing $\delta_x$ along each pixel location. Using gradient descent with learning rate $\gamma$, we update only the pixels within the trigger-pixel placements $E_t$ and assign the new values to $V_t$. In subsequent iterations, we replace the trigger-pixels in each image with their corresponding values from $V_t$, ensuring that only the trigger-pixels affect the loss.

## 4.4. Poisoning Malicious Clients' Training Data

The last step of our attack is to poison malicious clients' local training data using the optimized trigger $\tau = (E_t, V_t)$ and its target label $y_t$ by a certain data poison rate.

## 5. Theoretical Analysis

Gounding in the feature learning propertires of neural networks (Shi et al., 2022; Zeiler, 2014; Girshick et al., 2014), we assume a dataset $D$'s valid information can be extracted as a feature set, expressed as $K = (v_1, v_2, \ldots, v_k) \in \mathbb{R}^{n \times k}$. Each $v_i \in \mathbb{R}^n$ has a target value $y_i$, and $y = (y_1, y_2, \ldots, y_k) \in \mathbb{R}^k$. For a linear system $w \in \mathbb{R}^{n \times 1}$, the learning objective is to find $w^* \in \mathbb{R}^n$ such that $K^T w^* = y^T$. See proof of 5.1 in Appendix B.1.

**Proposition 5.1.** *(Concealment Property) Given a feature set $K \in \mathbb{R}^{n \times k}$ with its target values $y \in \mathbb{R}^k$ and a model $w \in \mathbb{R}^n$, assume an adversary generates a malicious feature set $K_{adv} \in \mathbb{R}^{n \times p}$ with adversarial target values $y_{adv} \in \mathbb{R}^p$. Let the error of $(K_{adv}, y_{adv})$ on $w$ be denoted as $\epsilon_{adv}$, where $\epsilon_{adv} = K_{adv}^T w - y_{adv}^T$. Let the optimization direction for $w$ with respect to $(K, y)$ be denoted by $\Delta w_K$, and the optimization direction for $w$ with respect to the combined feature set $([K, K_{adv}], [y, y_{adv}])$ be denoted by $\Delta w_{K \cup K_{adv}}$. The difference between the two update directions is bounded as:*

$$\|\Delta w_{K \cup K_{adv}} - \Delta w_K\| \leq \delta \|\epsilon_{adv}\|$$

*where $\delta = \max \|v_i\|, v_i \in K_{adv}$, representing the maximum magnitude of the feature vectors in the adversarial dataset $K_{adv}$. Specifically, this bound indicates that the difference between the two update directions is proportional to the error in optimizing $(K_{adv}, y_{adv})$ for $w$.*

## 6. Experiments

### 6.1. FL Configurations

We conducted experiments on four benchmark image datasets: Fashion MNIST, FEMNIST, CIFAR10, and Tiny ImageNet, using four different model architectures including ResNet and VGGNet, as detailed in Table 6. For FL settings, we consider 100 clients for grayscale image learning tasks and 50 clients for colorful image learning tasks. Clients' data are Non-iid distributed, where the Non-iid sampling followed the algorithm proposed by FLTrust (Cao et al., 2021), with a medium bias degree of 0.5. FEMNIST is naturally a Non-iid distributed dataset for FL, so we used it as is. Each client performed five local training epochs per global round and participated in all global rounds.

For grayscale image learning tasks, we used a fixed local learning rate of 0.1. For color image learning tasks, we applied learning rate scheduling techniques (He et al., 2016; Simonyan & Zisserman, 2015). We used SGD optimization

with CrossEntropy loss. In the experiments on Tiny ImageNet, we set the mini-batch size to 64, while for the other datasets, we set it to 256. The number of global rounds was determined based on the stabilization of test accuracy on the main task data, defined as remaining within 0.5 percentage points over five consecutive global rounds, which we considered convergence. The number of global rounds varied across datasets and model architectures, as detailed in Table 7.

### 6.2. Attack Configurations

The default Malicious Client Ratio (MCR) was set to 5%, meaning 2 out of 50 or 5 out of 100 clients engaged in data poisoning during training. The Data Poison Rate (DPR), representing the proportion of each malicious client's data poisoned with the $DPOT_{L_0}$ trigger, was set to a default value of 0.5. Except for ablation studies or specific indications, all experiments followed the default configurations for MCR and DPR.

### 6.3. Evaluation Metrics

We used Final Attack Success Rate (Final $ASR$) and Average Attack Success Rate (Avg $ASR$) to evaluate the effectiveness of backdoor attacks in FL. Final $ASR$, calculated as the mean $ASR$ of the global model from the last five rounds, measures the final effectiveness of the attack. Avg $ASR$, calculated as the mean $ASR$ across all global rounds, assesses the average by-round effectiveness. A higher Avg $ASR$ indicates faster achievement of sufficient attack effectiveness.

We used Main-task Accuracy ($MA$) to evaluate the performance of the final global model on main-task data. A backdoor attack is considered to maintain main-task convergence if the $MA$ of its victim model is within a $\pm 2$ percentage-point difference compared to the $MA$ of the un-attacked model.

### 6.4. Other Backdoor Triggers

In this study, we consider data poisoning as the sole attack strategy for backdoor attacks in FL. Similar attack settings in existing literature are relatively scarce, with many current FL attacks combining data poisoning with other strategies or targeting novel FL structures. Rather than comparing all existing attack methods across various configurations, we choose to compare representative backdoor triggers with $DPOT_{L_0}$'s trigger in a purely data poisoning attack context under unified settings to evaluate their effectiveness.

- **Fixed Trigger (FT).** A single pixel-pattern trigger with fixed value, shape, and placement (Baruch et al., 2019; Bagdasaryan et al., 2020).
- **Distributed Fixed Trigger (DFT).** Different fixed triggers are used by malicious clients, with their union employed for testing (Xie et al., 2020).

- **Partially $L_0$-norm-bounded Optimized Trigger ($OT_{L_0}^{val}$).** A single pixel-pattern trigger with dynamically optimized values but fixed shape and placement (Zhang et al., 2024).
- **$L_2$-norm-bounded Optimized Trigger ($OT_{L_2}$).** Adversarial perturbations added on data, generated with constraints of their $L_2$-norm (Nguyen et al., 2024).

### 6.5. Defenses

We selected defenses that have open-sourced their proof-of-concept code to ensure accurate implementation of their proposed ideas. Twelve of them are state-of-the-art server-conduct defenses based on analyzing difference of model updates from clients: FedAvg (McMahan et al., 2017), Median (Yin et al., 2018), Trimmed Mean (Yin et al., 2018), RobustLR (Ozdayi et al., 2021), RFA (Pillutla et al., 2022), FLAIR (Sharma et al., 2023), FLCert (Cao et al., 2022), FLAME (Nguyen et al., 2022), FoolsGold (Fung et al., 2020), Multi-Krum (Blanchard et al., 2017), BackdoorIndicator (Li & Dai, 2024), and FRL (Mozaffari et al., 2023). Detailed descriptions can be found in Appendix D. One defense is conducted on client-side: Flip (Zhang et al., 2023). Experiment results of Flip, FRL, and BackdoorIndicator are given in Appendix G, H, and I due to space limitations.

### 6.6. $DPOT_{L_0}$ vs. $OT_{L_2}$ vs. $OT_{L_0}^{val}$

We present $OT_{L_2}$, $OT_{L_0}^{val}$, and $DPOT_{L_0}$ triggers on CIFAR10 images in Figure 3. The size of the $DPOT_{L_0}$ trigger is set to 25, based on the subtlety maintenance rule. The $OT_{L_0}^{val}$ trigger consists of 25 pixels arranged in a square shape. We placed it in two different positions in the images - upper-left ($OT_{L_0}^{val}$-1) and center ($OT_{L_2}$-2 ) . Both $OT_{L_2}$ and $OT_{L_0}^{val}$ triggers are optimized to minimize backdoor loss on each round's global model before being used for training. Their optimization methods are based on two recent attack works: A3FL (Zhang et al., 2024) and IBA (Nguyen et al., 2024).

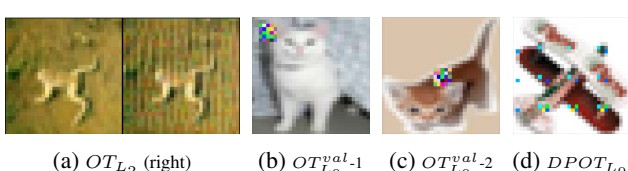

(a) $OT_{L_2}$ (right)    (b) $OT_{L_0}^{val}$-1    (c) $OT_{L_0}^{val}$-2    (d) $DPOT_{L_0}$

*Figure 3.* Different triggers on CIFAR10 images.

The comparative results of the three optimized triggers in terms of Final $ASR$, Avg $ASR$, and $MA$ are presented in Table 1. Compared to the $OT_{L_0}^{val}$ and $DPOT_{L_0}$ triggers, the $OT_{L_2}$ trigger demonstrates lower attack effectiveness in both final and average $ASR$. A potential explaination is that when the MCR is small (5%), the global model's updates are largely irrelevant to learning backdoor features, which impacts $OT_{L_2}$ triggers more than the $L_0$-norm bounded trig-

gers, as the $OT_{L_2}$ trigger contains more adversarial features to learn. Moreover, the accuracy of poisoned data with a larger number of features may be more negatively impacted by irrelevant changes in the global model along training. The enhanced attack effectiveness of $OT_{L_0}^{val}$-2 compared to $OT_{L_0}^{val}$-1 underscores the significance of trigger placement as a key factor to achieve backdoor attack objectives in FL. The $DPOT_{L_0}$ trigger with placement optimization therefore shows best effectiveness among all triggers.

The baseline MA results of an un-attacked FL system employed with various defense strategies are shown in the CIFAR10 column of Table 8. The MA results in Table 1 show minimal differences from the corresponding baseline values, indicating that attacks with different triggers maintain the main-task convergence of the FL system.

*Table 1.* Results of $OT_{L_2}$, $OT_{L_0}^{val}$, and $DPOT_{L_0}$ on CIFAR10.

| Measures | Final ASR | | | | Avg ASR | | | | MA | | | |
|---|---|---|---|---|---|---|---|---|---|---|---|---|
| Trigger Types | $OT_{L_2}$ | $OT_{L_0}^{val}$-1 | $OT_{L_0}^{val}$-2 | $DPOT_{L_0}$ | $OT_{L_2}$ | $OT_{L_0}^{val}$-1 | $OT_{L_0}^{val}$-2 | $DPOT_{L_0}$ | $OT_{L_2}$ | $OT_{L_0}^{val}$-1 | $OT_{L_0}^{val}$-2 | $DPOT_{L_0}$ |
| FedAvg | 18.5 | 48.9 | 75.1 | **100** | 26.0 | 38.1 | 60.2 | **98.5** | 69.3 | 70.6 | 70.0 | 70.7 |
| Median | 21.8 | 32.9 | 28.4 | **100** | 14.2 | 24.0 | 26.7 | **96.1** | 69.8 | 69.1 | 69.9 | 69.1 |
| TrimmedMean | 10.2 | 35.0 | 85.5 | **100** | 12.2 | 23.5 | 62.8 | **88.6** | 69.7 | 69.9 | 70.2 | 70.4 |
| RobustLR | 32.8 | 46.2 | 86.5 | **100** | 33.7 | 40.7 | 65.6 | **98.6** | 70.3 | 71.2 | 70.3 | 70.1 |
| RFA | 9.0 | 24.7 | 41.6 | **100** | 9.5 | 23.8 | 38.8 | **97.8** | 70.4 | 70.2 | 70.7 | 70.7 |
| FLAIR | 0.1 | 13.2 | 14.9 | **62.3** | 3.7 | 12.5 | 17.0 | **50.7** | 70.5 | 70.7 | 69.2 | 70.6 |
| FLAME | 3.9 | 13.7 | 18.2 | **59.8** | 25.1 | 32.1 | 48.4 | **56.0** | 68.7 | 70.1 | 69.5 | 70.3 |
| FoolsGold | 14.8 | 46.9 | 64.8 | **100** | 13.4 | 38.0 | 50.8 | **98.5** | 70.1 | 70.8 | 70.5 | 71.0 |
| FLCert | 4.0 | 39.0 | 28.9 | **99.2** | 4.2 | 28.4 | 25.5 | **88.3** | 69.3 | 69.9 | 69.7 | 70.0 |
| Multi-Krum | 0.3 | 33.4 | 86.2 | **100** | 4.4 | 29.5 | 84.2 | **98.7** | 64.3 | 62.8 | 61.2 | 63.0 |

### 6.6.1. MAIN-TASK CONVERGENCE BY $OT_{L_2}$

To study the long-term impact of the $OT_{L_2}$ trigger on main-task convergence in FL, we set the MCR to 50% to boost its attack effectiveness. The trigger was optimized for 30 rounds, and the poisoned data generated in the 30th round was used until the end (150th round). This simulates an unideal scenario where the attack is interrupted mid-training, allowing us to assess how remaining poisoned data affects MA over time. The results of $DPOT_{L_0}$ under identical conditions are presented in Table 2 for comparison.

*Table 2.* Insufficient MA due to $OT_{L_2}$ attack.

| | FedAvg | | | FLAIR | | |
|---|---|---|---|---|---|---|
| | Final ASR | Avg ASR | MA | Final ASR | Avg ASR | MA |
| $OT_{L_2}$ | 64.4 | 60.5 | **60.0** | 57.4 | 60.1 | **60.7** |
| $DPOT_{L_0}$ | 95.2 | 96.5 | **69.9** | 74.5 | 82.8 | **70.6** |

As we analyzed before, substantially changing benign features to adversarial features makes $OT_{L_2}$ overdetermine the learning objective and hinder the convergence of main-task data. In contrast, $DPOT_{L_0}$ is able to sustain the MA of FL training, even under conditions of a large MCR and an interruption in optimization.

### 6.6.2. COMPUTATIONAL OVERHEAD COMPARISON

We compared the elapsed time of trigger optimization algorithms in A3FL and IBA with $DPOT_{L_0}$ on the same computational platform, consisting of one NVIDIA A40 GPU core and 200 GB of CPU RAM. Comparison results are shown in Table 3. $DPOT_{L_0}$ demonstrates a relatively shorter total execution time. We assume adversaries can offset the timing gap caused by trigger optimization with powerful computational resources.

*Table 3.* Comparison of Elapsed Time

| Methods | Total (s) | Per Epoch (s) | # Epochs | Benign Training (s) |
|---|---|---|---|---|
| $DPOT_{L_0}$ | **5.05** | **0.50** | 10 | 1.23 |
| A3FL | 421.04 | 2.07 | 200 | 1.23 |
| IBA | 16.56 | 1.59 | 10 | 1.23 |

### 6.6.3. SUBTLETY COMPARISON.

We evaluated the subtlety of four optimized triggers by measuring the accuracy drop of an un-attacked model when predicting poisoned data as benign labels ("Benign Acc" in Table 4). $OT_{L_2}$ showed a relatively greater drop in benign accuracy due to its substantial alteration of benign features across the entire image.

*Table 4.* Benign accuracy drops caused by different triggers.

| Triggers | None | $DPOT_{L_0}$ | $OT_{L_0}^{val}$-1 | $OT_{L_0}^{val}$-2 | $OT_{L_2}$ |
|---|---|---|---|---|---|
| Benign acc | 70.81 | 52.98 | 70.46 | 67.65 | **27.98** |
| Drop (%) | 0 | 25.18 | 0.49 | 4.46 | **60.49** |

### 6.7. $DPOT_{L_0}$ vs. FT vs. DFT

Figure 4 presents a comparison of the $ASR$ results for the $DPOT_{L_0}$ trigger, FT, and DFT across different datasets. Visualizations of FT and DFT can be found in Figures 8 and 9, respectively. The MA results are provided in Table 8.

### 6.8. Discussion of $DPOT_{L_0}$

#### 6.8.1. AGGREGATION OF MALICIOUS MODEL UPDATES

We demonstrated that $DPOT_{L_0}$'s attack effectiveness arises from malicious model updates being aggregated into the global model, rather than solely from the optimized trigger's residual effects on the next-round global model.

To evaluate this, we designed an experiment where malicious clients generated a $DPOT_{L_0}$ trigger every round but did not use it to poison their data. We tested the $ASR$ of the trigger on the next-round global model, measuring its residual effects, and denoted this as $\widetilde{ASR}$. In another experiment, malicious clients input the poisoned data with the $DPOT_{L_0}$ trigger into the training, with the attack effectiveness denoted as $\ddot{ASR}$.

As shown in Appendix Table 9, $\ddot{ASR}$ is notably larger than $\widetilde{ASR}$ under different defenses across various datasets. These results validate that malicious model updates can effectively bypass defenses, be aggregated into the global model, and drive it into a backdoored state.

#### 6.8.2. WORKING PRINCIPLE ANALYSIS

The working principle of $DPOT_{L_0}$ in backdoor attack can be explained through the relationship between its $ASR$ and

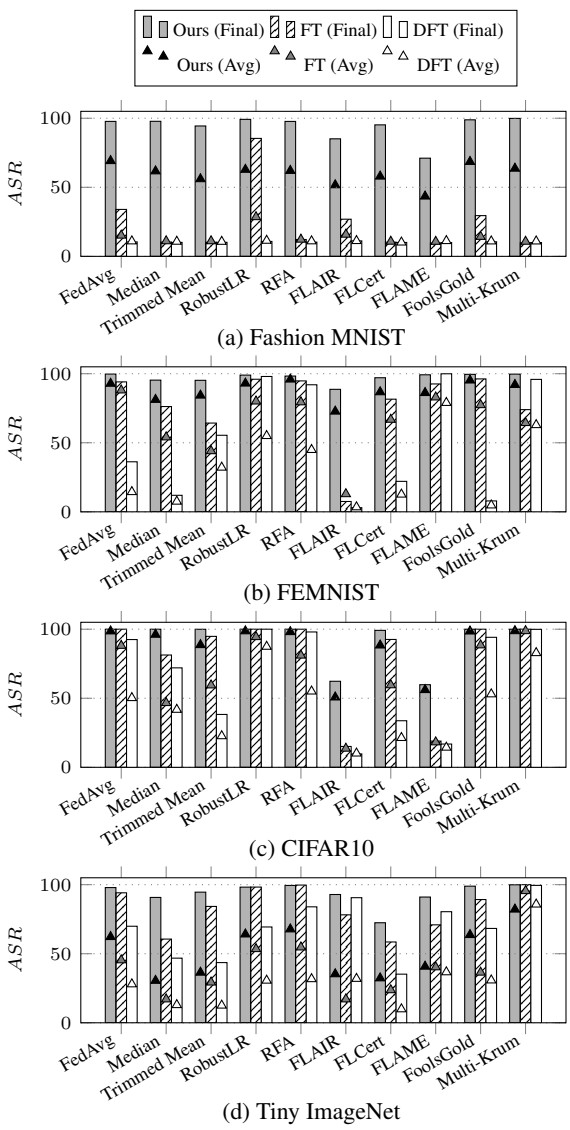

*Figure 4.* $ASR$ results for the $DPOT_{L_0}$, FT, and DFT.

the duration of the attack. We conducted experiments where the attack was initiated at different training rounds, and $ASR$ was observed at specific subsequent rounds.

Table 5a presents the results for Fashion MNIST with Trimmed Mean as the defense strategy. The $ASR$ increases with training duration and prior attack presence. This shows the global model gradually learns the backdoor feature. The optimization of the $DPOT_{L_0}$ trigger in this scenario enables malicious model updates to bypass the defense mechanism and accelerates the global model's learning of the backdoor feature. This is achieved through the alternating optimization of the model and trigger, both aimed at minimizing backdoor loss. The final $ASR$ results in this case are better than using $DPOT_{L_0}$ algorithms to perform adversarial attack on an un-attacked model.

Table 5b shows the results for CIFAR10 with FLAIR as

the defense strategy. The $ASR$ exhibits little variation regardless of attack duration or the presence of prior attacks. This indicates that the malicious model updates are weakly bypass the defense under current attacking pattern. Consequently, the $ASR$ primarily reflects the residual effects of the trigger on the next-round global model. We found a small difference between the $ASR$ by adversarial and backdoor attacks, indicating that the $DPOT_{L_0}$ trigger, with limited backdoor features, has good transferability across rounds. A more effective backdoor attacking pattern for this case can be found in Appendix P.

In summary, $DPOT_{L_0}$ combines pixel-pattern triggers' learnability with adversarial triggers' transferability, demonstrating varied efficacy across conditions.

*Table 5.* $DPOT_{L_0}$ for backdoor attack and for adversarial attack.

(a) $ASR$ is dependent to backdoor attack duration.

| | | Backdoor Attack | | | | | Adversarial Attack |
|---|---|---|---|---|---|---|---|
| | | Observe at (round): | | | | | |
| | | 1 | 200 | 250 | 280 | 300 | |
| Attack starts at (round): | 1 | 10.0 | 76.27 | 89.52 | 93.6 | 95.64 | 48.92 |
| | 200 | - | 49.56 | 84.03 | 91.26 | 93.14 | |
| | 250 | - | - | 69.47 | 81.04 | 87.86 | |
| | 280 | - | - | - | 66.75 | 74.41 | |

(b) $ASR$ is independent to backdoor attack duration.

| | | Backdoor Attack | | | | | Adversarial Attack |
|---|---|---|---|---|---|---|---|
| | | Observe at (round): | | | | | |
| | | 1 | 100 | 140 | 145 | 150 | |
| Attack starts at (round): | 1 | 10.0 | 57.48 | 65.43 | 60.72 | 61.29 | 56.79 |
| | 100 | - | 47.54 | 74.95 | 64.04 | 61.14 | |
| | 140 | - | - | 62.62 | 57.19 | 59.11 | |
| | 145 | - | - | - | 63.63 | 63.42 | |

### 6.8.3. MORE RESULTS

Additional results of potential interest to readers are provided in the Appendix. Section J presents experimental evidence guiding trigger size selection for different datasets. The evolution of the $DPOT_{L_0}$ trigger during FL training is visualized in Section K. Ablation studies on the effects of different MCR, trigger size, DPR, Non-iid degree, and attacking patterns on $DPOT_{L_0}$'s performance are detailed in Sections L, M, N, O, and P, respectively. We also discussed the attack performance of combining $DPOT_{L_0}$ with model-poisoning techniques by relaxing the TEEs constraints in Section Q.

## 7. Conclusion

In this work, we proposed $DPOT_{L_0}$, a novel backdoor attack method relying solely on data poisoning in Federated Learning (FL). $DPOT_{L_0}$ dynamically adjusts the backdoor objective to conceal malicious clients' model updates among benign ones, enabling global models to aggregate them even when protected by state-of-the-art defenses.

## Impact Statement

**Ethics Statement:** This paper presents work whose goal is to advance the field of Machine Learning. Our paper presents a practical attack on federated learning, which can be executed with minimal technical skill by anyone who can participant into an FL. While this may seem risky, we believe the benefits of disclosing this attack outweigh potential harms. Sharing the limitations of current defense strategies early prevents future misuse in security-critical applications, allowing organizations to address vulnerabilities before widespread deployment.

**Reproducibility Statement:** To ensure the reproducibility of our results, we have provided detailed descriptions of our experimental setup, including model architectures, hyperparameters, datasets, and training procedures. All code used to implement our attack and run evaluations will be made available after the publication of this paper. Additionally, our code can be easily adapted to other FL research projects by simply integrating our algorithms into the data preparation process of FL clients before the data is input into their training phase. Therefore, our work can be extensively used to evaluate future FL systems for security purposes.

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

# A. Additional Related Works

### A.1. Federated Learning (FL)

The Federated Learning (McMahan et al., 2017) (FL) training process involves four main steps: 1) **Model Distribution**: A central server distributes the most recent global model to the participating clients. 2) **Local Training**: Each client independently trains the global model on its local training dataset and obtains a local model. 3) **Model Updates**: Each client calculates the parameter-wise difference between its local model and the global model, referred to as model updates, and then sends them to the central server. 4) **Aggregation**: The central server aggregates clients' model updates to create a new global model. This entire process, consisting of step 1 to 4, constitutes a global round. The FL system repeats these steps for a certain number of rounds to obtain a final version of the global model.

### A.2. Backdoor Attacks in FL

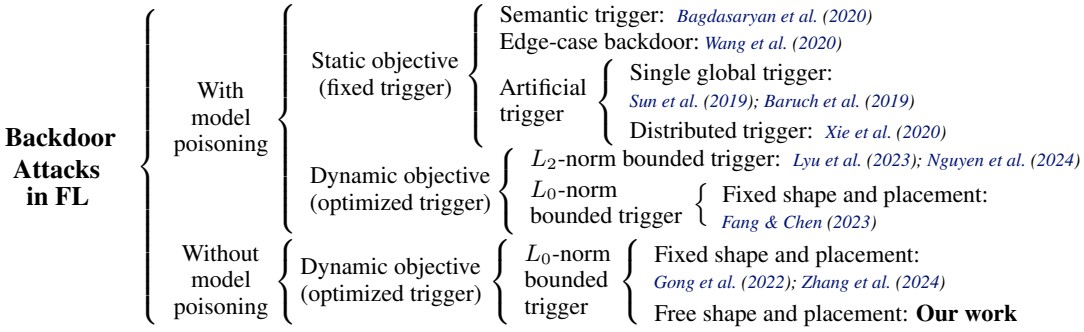

*Figure 5.* An overview of related works on backdoor attacks in FL.

FL is easily suffered from backdoor attacks. As training data are privately held by clients, the security of data is hard to track or protect. Adversaries can inject backdoors into the global model simply by compromising a few vulnerable client devices and poisoning their data with backdoor triggers. To date, many variations of backdoor attacks targeting FL have emerged, and we summarize those specific to image classification tasks in Figure 5.

**With model poisoning vs. Without model poisoning**

The foundation of backdoor attacks in FL is through ***data poisoning*** - attackers embed backdoor triggers into the local training data of certain clients and change the ground-truth labels of the infected data to malicious labels. As a result, clients' local models trained on the poisoned data will be backdoored, and consequently, the global model that aggregates these backdoored models will also be backdoored.

A standalone data poisoning is found challenging to succeed when employing some types of triggers. Therefore, many works introduce model poisoning to assist backdoor attacks in FL. ***Model poisoning*** aims to either directly manipulate clients' model updates or indirectly achieve this by changing their local training algorithms. Three main approaches in model poisoning were widely adopted in existing attacks: 1) Scaling based (Bagdasaryan et al., 2020; Sun et al., 2019; Xie et al., 2020; Gong et al., 2022). Attackers amplify malicious model updates generated from backdoored models before clients send them to the server. These malicious updates can overpower the aggregation results, causing the global model to quickly incorporate backdoors. However, this approach is vulnerable to defenses that exclude outlier model updates from the aggregation. 2) Constraint based (Bagdasaryan et al., 2020; Lyu et al., 2023). Attackers change clients' local training algorithms by adding extra constraints to their loss functions, giving backdoored models specific characteristics, such as being less distinguishable from benign models. 3) Projection based (Zhang et al., 2022; Baruch et al., 2019; Wang et al., 2020; Fang & Chen, 2023). Attackers constrain backdoor implementation to bounded model parameters: by clipping parameter values or using Projected Gradient Descent, backdoor models are $L_2$-norm bounded to a chosen model state; by selectively updating a subset of parameters, they are $L_0$-norm bounded to a chosen state.

Model poisoning requires attackers to modify certain clients' local training procedures. However, with the introduction of Trusted Execution Environments (TEEs) by state-of-the-art defense mechanisms (Riege et al., 2024), client-side execution for training can be authenticated and secure, thus increasing the difficulty of conducting model poisoning. In contrast, data poisoning is easier to conduct and harder to prevent since clients may collect their local data from open resources where attackers can also get access to and make modifications.

**Static objective vs. Dynamic objective**

If a backdoor attack has a specified and unchanging objective that is independent to the training system's status, we refer to this as a ***static objective***. For instance, Semantic trigger as backdoor (Bagdasaryan et al., 2020) aims to associate certain features from input that is unrelated to the main training tasks with an attacker-chosen output, causing the model to make incorrect predictions on those inputs; Edge-case backdoor (Wang et al., 2020) selects data that share certain commonalities but are from the tail end of the input data distribution as the backdoored input, causing the model to mispredict them; Artificial trigger as backdoor (Sun et al., 2019; Zhang et al., 2022; Baruch et al., 2019; Xie et al., 2020) embeds a few pixels forming a specific artificial pattern into the input, leading the model to mispredict any input containing this pixel pattern. In FL, since the static objectives of backdoor attacks are inconsistent with the optimization objectives defined by the main-task data, malicious models will exhibit distinct differences in their model updates compared to benign models, making them easy to detect.

In contrast to a static objective, a backdoor attack that adjusts its objective based on the training system's status is referred to as having a ***dynamic objective***. By adjusting its objective, a backdoor attack is expected to achieve greater effectiveness. Several approaches have been proposed in recent attack studies to attempt to accomplish this. For example, Model-dependent attack (Gong et al., 2022) and F3BA (Fang & Chen, 2023) optimized the trigger pattern based on a hypothesis that maximizing the activation of certain neurons in the backdoored local model can enhance the attack's persistence on the global model, which provides preliminary insights into the potential of optimized triggers; A3FL (Zhang et al., 2024), which optimizes triggers specifically for a defense scenario where the global model is directly trained to unlearn the trigger, is another pioneering work exploring the potential of optimized triggers in attacking FL.

**$L_2$-norm-bounded optimized trigger vs. $L_0$-norm-bounded optimized trigger**

A critical consideration in designing backdoor triggers is ensuring their subtlety when applied to input data, resulting in a trivial disparity between human perception and the backdoored model's interpretation. Existing dynamic objective attacks achieve this by constraining the optimized triggers' $L_2$-norm or $L_0$-norm bounds.

An $L_2$-***norm-bound*** restricts the total magnitude of the perterbations adding to the data. For example, CerP (Lyu et al., 2023) and IBA (Nguyen et al., 2024) generate optimized perturbations adds them to clients' local data to induce their local models learn to misclassify the perturbed data to a specified target label.

An $L_0$-***norm bound*** restricts the number of components (e.g., pixels in an image) that can be altered by the trigger. For

example, The optimized trigger in Model-dependent attack (Gong et al., 2022), F3BA (Fang & Chen, 2023), or A3FL (Zhang et al., 2024) consists of a small number of pixels arranged in a square shape, being placed in a certain location on the data.

**Clean-label attacks**

Clean-label attacks (Shafahi et al., 2018) involve manipulating input data with subtle perturbations while keeping labels unchanged. Although this assumption aligns with scenarios like Vertical Federated Learning (Liu et al., 2024) (VFL), where participants possess vertically partitioned data with labels owned by only one participant, our study does not consider VFL as our attack scenario. Therefore, discussions of clean-label attacks are beyond the scope of our work.

### A.3. Defenses with different privacy-preserving properties

Recent defense works have introduced several novel FL pipelines aimed at enhancing the security of FL against various types of attacks. These novel architectures provide different levels of privacy protection and often require additional techniques (e.g., Secured Multi-party Computation) to ensure privacy for FL clients. In light of these privacy considerations, we have chosen to focus our analysis on the conventional FL structure that was originally proposed in the concept of Federated Learning (McMahan et al., 2017). Although defenses built on newly proposed FL structures fall outside the scope of our main comparison, we offer a discussion of these related works in this section.

**Clients' private data were shared to the server:** Some approaches allow the server to have access to a small portion of main-task data shared by clients. To mitigate backdoor attacks, server-side defense strategies use this data to either independently train a model and use its updates as a reference for each round of aggregation (e.g., FLTrust (Cao et al., 2021)), or to validate clients' model updates and eliminate those with abnormal outputs (e.g., SSDT (Mo et al., 2024), SHERPA (Sandeepa et al., 2024)). However, both of these methods still rely on analyzing clients' model updates, making them vulnerable to backdoor attacks with dynamic objectives that conceal malicious updates. FedREdefense (Xie et al., 2024) detects and filters out artificial model updates by reconstructing distilled data shared by clients, but this approach is not effective against backdoor attacks where malicious clients genuinely train their models on poisoned local data rather than fabricating model updates.

**Clients' model updates were shared to each other:** Some approaches propose allowing clients to share their model updates with one another, rather than just with the server. CrowdGuard (Riege et al., 2024) and FLShield (Kabir et al., 2024) suggest that a subset of clients validate other clients' model updates using their own data, assuming that malicious model updates would produce abnormal outputs on benign data. However, this hypothesis fails when malicious model updates are trivially different from non-attacked model updates, a state that can be achieved through using optimized triggers. Fang et al. (2024) proposed a decentralized FL framework without a central server, where clients exchange model updates and apply Byzantine-robust aggregation using their own updates as a reference. Like other defenses that rely on analyzing clients' model updates, this approach is also vulnerable to backdoor attacks with optimized triggers.

## B. Theoretical Analysis

### B.1. Proof of Proposition 5.1

*Proof.* We define the least-squares optimization objectives for $f_K$ and $f_{K \cup K_{adv}}$:

$$f_K = \frac{1}{2}\|K^T w - y^T\|_2^2 \tag{1}$$

$$f_{K \cup K_{adv}} = \frac{1}{2}\|[K \ K_{adv}]^T w - [y \ y_{adv}]^T\|_2^2. \tag{2}$$

The gradients with respect to $w$ are:

$$\frac{\partial f_K}{\partial w} = K(K^T w - y^T),$$

$$\frac{\partial f_{K \cup K_{adv}}}{\partial w} = [K \ K_{adv}]([K \ K_{adv}]^T w - [y \ y_{adv}]^T)$$

$$= (KK^T + K_{adv}K_{adv}^T)w - (Ky^T + K_{adv}y_{adv}^T)$$

$$= K(K^T w - y^T) + K_{adv}(K_{adv}^T w - y_{adv}^T).$$

Let $\epsilon_{adv}$ represent the error of $(K_{adv}, y_{adv})$ on the model $w$:

$$\epsilon_{adv} = K_{adv}^T w - y_{adv}^T.$$

Substituting $\epsilon_{adv}$ into the gradient, we get:

$$\frac{\partial f_{K \cup K_{adv}}}{\partial w} = K(K^T w - y^T) + K_{adv}\epsilon_{adv}.$$

The difference in gradients is:

$$\Delta = \frac{\partial f_{K \cup K_{adv}}}{\partial w} - \frac{\partial f_K}{\partial w} = K_{adv}\epsilon_{adv}. \tag{3}$$

Writing $\epsilon_{adv} \in \mathbb{R}^p$ as $\epsilon_{adv} = (e_1, e_2, \ldots e_p)$ and $K_{adv} \in \mathbb{R}^{n \times p}$ as $K_{adv} = (v_1, v_2, \ldots, v_p)$, where $v_i \in \mathbb{R}^n$, the magnitude of $\Delta$ is bounded as:

$$\|\Delta\| = \|K_{adv}\epsilon_{adv}\| = \|e_1 v_1 + e_2 v_2 + \cdots + e_p v_p\| \leq \sum_{i=1}^{p} \|e_i v_i\| \leq \delta \|\epsilon_{adv}\|,$$

where $\delta = \max \|v_i\|, v_i \in K_{adv}$.

Finally, the update directions $\Delta w_K$ and $\Delta w_{K \cup K_{adv}}$ for minimizing the objective 1 and 2, defined as the negative gradients, satisfy:

$$\|\Delta w_{K \cup K_{adv}} - \Delta w_K\| = \|\Delta\| \leq \delta \|\epsilon_{adv}\|.$$

Thus, when $\epsilon_{adv} = 0$, the update directions for $f_K$ and $f_{K \cup K_{adv}}$ are identical. Otherwise, the difference is bounded by $\delta \|\epsilon_{adv}\|$, quantifying the influence of the adversarial error.

$\square$

## C. Experimental Settings

Table 6. Dataset description

| Dataset | #class | #img | img size | Model | #params |
|---|---|---|---|---|---|
| Fashion MNIST | 10 | 70k | $28 \times 28$ grayscale | 2 conv 3 fc | $\sim$1.5M |
| FEMNIST | 62 | 33k | $28 \times 28$ grayscale | 2 conv 2 fc | $\sim$6.6M |
| CIFAR10 | 10 | 60k | $32 \times 32$ color | ResNet18 | $\sim$11M |
| Tiny ImageNet | 200 | 100k | $64 \times 64$ color | VGG11 | $\sim$35M |

## D. Descriptions of Defenses

Twelve different server-side defense strategies, based on analyzing clients' model updates, are briefly introduced below:

**FedAvg** (McMahan et al., 2017), a basic aggregation rule in FL, computes global model updates by averaging all clients' model updates.

*Table 7.* Default settings

| | Trigger Size | Round | Number of Clients | Malicious Client Ratio | Data Poison Rate |
|---|---|---|---|---|---|
| Fashion MNIST | 64 | 300 | 100 | | |
| FEMNIST | 25 | 200 | 100 | | |
| CIFAR10 | 25 | 150 | 50 | 0.05 | 0.5 |
| Tiny ImageNet | 64 | 100 | 50 | | |

**Median (Yin et al., 2018)**, a simple but robust alternative to FedAvg, constructs the global model updates by taking the median of the values of model updates across all clients

**Trimmed Mean (Yin et al., 2018)**, in our implementation, excludes the $40\%$ largest and $40\%$ smallest values of each parameter among all clients' model updates and takes the mean of the remaining $20\%$ as the global model updates.

**Multi-Krum (Blanchard et al., 2017)**, in our implementation, identifies 10% honest client whose model updates have the smallest Euclidean distance to all other clients' model updates and takes the average of these honest clients' model updates as the global model updates.

**RobustLR (Ozdayi et al., 2021)** adjusts the aggregation server's learning rate, per dimension and per round, based on the sign information of clients' updates.

**RFA (Pillutla et al., 2022)** computes a geometric median of clients' model updates and assigns weight factors to clients depending on their distance from the geometric median. Subsequently, it computes the weighted average of all clients' model updates to generate the global model updates.

**FLAIR (Sharma et al., 2023)** assigns different weight factors to clients according to the similarity of the coefficient signs between client model updates and global model updates of the previous round, and then takes the weighted average of all clients' model updates to form the global model updates. The weight factors are carried over and accumulate from the previous round.

**FLCert (Cao et al., 2022)** randomly groups clients into 5 clusters, computes the median of model updates within each cluster, and uses the majority inference outcomes of these cluster models as the final results.

**FLAME (Nguyen et al., 2022)** first clusters clients' model updates according to their cosine similarity to each other, and then aggregates the clipped model updates within the largest cluster as the global model updates.

**FoolsGold (Fung et al., 2020)** reduces aggregation weights of a set of clients whose model updates constantly exhibit high cosine similarity to each other.

**BackdoorIndicator (Li & Dai, 2024)** trains an indicator model using OOD datasets to serve as the global model, then filtering out clients' model updates if their accuracy on those OOD datasets greater than a threshold.

**FRL (Mozaffari et al., 2023)** is a defense strategy where the server sparsifies the value space of model updates, allowing clients to vote on the most effective model updates based on their local data. The server then aggregates only the accepted votes while rejecting outliers to construct the global model.

## E. Main-task Accuracy Results corresponding to Figure 4

Table 8 lists the Main-task Accuracy of each experiment in getting results in Figure 4. Table 8 demonstrates that for different datasets used as the main tasks, global models under various attacks maintained a comparable level of Main-task Accuracy to the baselines with no attacks ("None"), indicating that all types of backdoor attacks successfully achieved their main-task convergence goals.

## F. Aggregation of Malicious Model Updates

In this section, we analyzed the attack effectiveness of each component of the $DPOT_{L_0}$ attack's working principles and report evidence that it effectively conceals malicious clients' model updates, thereby getting them integrated into the global models through aggregation.

*Table 8.* The Main-task Accuracies (MA) correspond to results in Figure 4. "None" represents no attack existing in the FL training.

| MA | Tiny ImageNet | | | | Fashion MNIST | | | | FEMNIST | | | | CIFAR10 | | | |
| --- | --- | --- | --- | --- | --- | --- | --- | --- | --- | --- | --- | --- | --- | --- | --- | --- |
| | None | Ours | FT | DFT | None | Ours | FT | DFT | None | Ours | FT | DFT | None | Ours | FT | DFT |
| FedAvg | 43.9 | 43.5 | 43.0 | 43.3 | 86.7 | 87.3 | 86.7 | 86.8 | 82.2 | 81.4 | 83.3 | 82.3 | 70.3 | 70.7 | 70.4 | 71.4 |
| Median | 40.6 | 40.2 | 40.6 | 38.6 | 86.0 | 85.8 | 86.6 | 86.3 | 80.4 | 81.5 | 79.8 | 79.9 | 70.2 | 69.1 | 69.8 | 69.7 |
| Trimmed Mean | 40.8 | 40.4 | 40.1 | 40.6 | 86.4 | 85.8 | 86.4 | 86.3 | 80.2 | 81.7 | 81.3 | 81.2 | 69.4 | 70.4 | 70.2 | 70.8 |
| RobustLR | 44.1 | 42.7 | 42.9 | 43.2 | 86.5 | 86.8 | 86.6 | 86.9 | 81.8 | 82.5 | 81.9 | 82.6 | 70.4 | 70.1 | 70.3 | 70.5 |
| RFA | 43.6 | 43.0 | 43.0 | 43.0 | 86.4 | 86.0 | 87.1 | 87.1 | 83.0 | 80.7 | 81.0 | 80.8 | 70.4 | 70.7 | 70.3 | 70.8 |
| FLAIR | 43.6 | 42.6 | 41.8 | 42.1 | 86.1 | 84.9 | 85.2 | 84.4 | 81.5 | 80.7 | 80.6 | 79.7 | 70.3 | 70.6 | 71.0 | 70.4 |
| FLCert | 40.3 | 40.2 | 39.7 | 39.7 | 86.2 | 85.9 | 86.0 | 86.8 | 81.3 | 80.9 | 81.5 | 81.0 | 69.6 | 70.0 | 69.8 | 70.4 |
| FLAME | 29.9 | 28.7 | 29.2 | 28.9 | 86.4 | 86.4 | 86.4 | 86.7 | 81.8 | 80.2 | 80.7 | 81.0 | 70.1 | 70.3 | 70.9 | 70.9 |
| FoolsGold | 43.1 | 43.2 | 43.5 | 43.2 | 86.6 | 87.1 | 86.8 | 87.3 | 83.4 | 82.7 | 83.0 | 81.8 | 70.4 | 71.0 | 71.2 | 71.7 |
| Multi-Krum | 30.7 | 27.7 | 27.7 | 26.4 | 86.2 | 85.9 | 86.0 | 87.0 | 79.9 | 80.4 | 79.6 | 80.2 | 61.4 | 63.0 | 63.2 | 60.8 |

In the $i$-th round, $DPOT_{L_0}$ generates a trigger $\tau^{(i)}$ by optimizing its shape, placement and values to make the global model of this round $W_g^{(i)}$ achieve a maximum $ASR$. However, what we were truly interested in is its $ASR$ on the global model after the $i$-th round aggregation, which is the next-round global model denoted as $W_g^{(i+1)}$. The attack effectiveness of the trigger $\tau^{(i)}$ on the global model $W_g^{(i+1)}$ stems from two factors:

1. **Trigger Optimization**: Trigger optimization using $W_g^{(i)}$ results in an improvement of the trigger's $ASR$ on $W_g^{(i+1)}$ due to the small difference between $W_g^{(i+1)}$ and $W_g^{(i)}$.

2. **Aggregation of Backdoored Model Updates**: Model updates that were trained on data partially poisoned by $\tau^{(i)}$ exhibit small differences from those were trained on data without poisoning. Therefore, they bypassed defenses and made $W_g^{(i+1)}$ incorporate backdoored model parameters.

In the following, we explain how we designed experiments to study the impact of each factor, and analyzed the experiment results.

**Experiment design:** To assess the attack effectiveness solely brought by Trigger Optimization, we eliminated any effects produced by data poisoning. Specifically, we set all clients in the FL system to be benign, ensuring that the next-round global model, denoted as $\widetilde{W}_g^{(i+1)}$, aggregated benign model updates only. In the meantime, we still collected data from a certain number of clients and optimized a trigger $\widetilde{\tau}^{(i)}$ for $\widetilde{W}_g^{(i)}$. Then, we tested $\widetilde{W}_g^{(i+1)}$ on a testing dataset in which all images are poisoned with the trigger $\widetilde{\tau}^{(i)}$ to obtain an $\widetilde{ASR}$. This $\widetilde{ASR}$ evaluates the attack effectiveness achieved by the current-round optimized trigger $\tau^{(i)}$ on the next-round global model $\widetilde{W}_g^{(i+1)}$, which does not contain any model updates learned from backdoor information.

To assess the attack effectiveness brought by Aggregation of Backdoored Model Updates, we introduced malicious clients into the FL system and therefore the global model, denoted as $\ddot{W}_g^{(i+1)}$, was allowed to aggregate model updates submitted by malicious clients. In this system, malicious clients partially poisoned their local training data (aligning with default settings in Table 7) using the trigger $\ddot{\tau}^{(i)}$ that was optimized for $\ddot{W}_g^{(i)}$, and then conducted their local training. We tested the $\ddot{W}_g^{(i+1)}$ on the testing dataset that was also poisoned by $\ddot{\tau}^{(i)}$ to obtain an $\ddot{ASR}$. We evaluated the attack effectiveness of Aggregation of Backdoored Model Updates by measuring the increase in $ASR$ compared to the previous setting, calculated as $(\ddot{ASR} - \widetilde{ASR})$. This metric reveals how much the malicious clients' model updates influenced the global model $\ddot{W}_g^{(i+1)}$ to achieve a higher $ASR$ compared to $\widetilde{W}_g^{(i+1)}$.

**Experiment results:** Table 9 shows results of $\widetilde{ASR}$ and $\ddot{ASR}$ over 10 different defense methods. We used same settings as in Table 7 for testing $\ddot{ASR}$, and kept the size of trigger training dataset consistent when testing $\widetilde{ASR}$.

The results of $\widetilde{ASR}$ in Table 9 show that different defense methods resulted in very different $\widetilde{ASR}$ even for the same learning task of a dataset. The reason for the variance of $\widetilde{ASR}$ is the gap between $W_g^{(i)}$ and $\widetilde{W}_g^{(i+1)}$ were different when implementing different defense methods. According to recent studies (Lyu et al., 2023; Zhang et al., 2024), if the gap between consecutive rounds of global models in an FL system is smaller, Trigger Optimization will be more effective in its attack. The results of $\ddot{ASR}$ in Table 9 show that the presence of malicious clients' model updates consistently enhances $ASR$ compared to $\widetilde{ASR}$ across all defense methods on different datasets. We consider this enhancement as an evidence of the statement that the attack effectiveness of $DPOT_{L_0}$ comes from both Trigger Optimization and Aggregation of Backdoored

*Table 9.* $ASR$ under different attacking conditions. $\widetilde{ASR}$ assesses the attack effectiveness of "Trigger Optimization" alone, while $A\ddot{S}R$ assesses the combined effectiveness of both "Trigger Optimization" and "Aggregation of Backdoored Model Updates".

| | $ASR$ type | Fashion MNIST Final | Fashion MNIST Avg | FEMNIST Final | FEMNIST Avg | CIFAR10 Final | CIFAR10 Avg |
|---|---|---|---|---|---|---|---|
| FedAvg | $\widetilde{ASR}$ | 58.8 | 45.1 | 54.0 | 28.6 | 55.6 | 50.9 |
| | $A\ddot{S}R$ | **97.7** | **69.1** | **99.7** | **92.9** | **100** | **98.5** |
| Median | $\widetilde{ASR}$ | 57.9 | 38.2 | 18.0 | 17.5 | 56.6 | 48.7 |
| | $A\ddot{S}R$ | **97.8** | **61.7** | **95.4** | **81.2** | **100** | **96.1** |
| Trimmed Mean | $\widetilde{ASR}$ | 31.6 | 29.7 | 24.2 | 25.6 | 55.6 | 40.9 |
| | $A\ddot{S}R$ | **94.4** | **56.0** | **95.2** | **84.3** | **100** | **88.6** |
| RobustLR | $\widetilde{ASR}$ | 70.2 | 47.2 | 28.8 | 27.3 | 60.1 | 47.3 |
| | $A\ddot{S}R$ | **99.2** | **62.8** | **99.3** | **93.0** | **100** | **98.6** |
| RFA | $\widetilde{ASR}$ | 78.0 | 46.4 | 18.9 | 13.4 | 57.4 | 46.1 |
| | $A\ddot{S}R$ | **97.7** | **62.0** | **98.3** | **95.9** | **100** | **97.8** |
| FLAIR | $\widetilde{ASR}$ | 42.2 | 36.2 | 23.0 | 29.6 | 54.1 | 45.9 |
| | $A\ddot{S}R$ | **85.3** | **50.1** | **88.7** | **72.7** | **62.3** | **50.7** |
| FLCert | $\widetilde{ASR}$ | 49.6 | 39.7 | 27.7 | 34.6 | 48.7 | 46.7 |
| | $A\ddot{S}R$ | **95.2** | **57.9** | **97.1** | **86.7** | **99.2** | **88.3** |
| FLAME | $\widetilde{ASR}$ | 38.0 | 26.2 | 34.7 | 35.7 | 28.1 | 51.0 |
| | $A\ddot{S}R$ | **71.1** | **43.4** | **99.2** | **86.1** | **59.8** | **56.1** |
| Fools Gold | $\widetilde{ASR}$ | 54.2 | 50.3 | 57.0 | 43.7 | 35.5 | 35.6 |
| | $A\ddot{S}R$ | **98.9** | **68.5** | **99.6** | **95.2** | **100** | **98.5** |
| Multi-Krum | $\widetilde{ASR}$ | 60.6 | 45.4 | 31.7 | 28.7 | 49.7 | 36.1 |
| | $A\ddot{S}R$ | **99.9** | **63.6** | **99.7** | **92.0** | **100** | **98.7** |

Model Updates, with the latter one playing a critical role in producing a high $A\ddot{S}R$.

A general hypothesis made by the state-of-the-art defenses against backdoor attacks in FL is that malicious clients' model updates have a distinct divergence from benign clients' model updates. However, as indicated by the results in Table 9, $DPOT_{L_0}$ effectively conceals the model updates from malicious clients amidst those of benign clients, eluding detection and filtering by state-of-the-art defenses. Consequently, defenses formulated based on this broad hypothesis will inherently struggle to defend against $DPOT_{L_0}$ attacks.

## G. Evaluation of $DPOT_{L_0}$ attack against Flip (Zhang et al., 2023)

Flip (Zhang et al., 2023) is a client-side defense strategy where benign clients perform trigger inversion and adversarial training using their local data to recover the global model from backdoors. In this section, we evaluate the effectiveness of the $DPOT_{L_0}$ attack against the Flip defense. We implemented the $DPOT_{L_0}$ attack by modifying the data preparation approach in Flip's open-source project, replacing it with the method used in this work, and injecting our data-poisoning algorithms into a subset of clients. Additionally, as $DPOT_{L_0}$ is a pure data-poisoning attack, we removed any additional steps in their project specified to malicious clients but not existed in benign clients' training, to ensure consistency between malicious clients and benign clients in FL training. We selected Fashion MNIST as the main-task dataset for our evaluation and directly adopted Flip's default experiment settings provided in their project - the total number of clients was 100 and 4% of them were malicious clients; the aggregation rule was set to FedAvg; the global model's parameters were initialized by a pre-trained state. The size of $DPOT_{L_0}$ trigger was set to 64, consistent with our default attacking settings.

We compared the performance of the $DPOT_{L_0}$ attack under two attack patterns provided by Flip's project: 1) **Single shot**: Each of the 4 malicious clients conducts a one-time attack at the beginning of training. 2) **Continuous**: All 4 malicious clients continuously execute the attack algorithms in every round during training.

Figure 6 shows the performance of the $DPOT_{L_0}$ attack on an FL system using Flip as its defense, measured by the Attack Success Rate (ASR). In the single-shot attack pattern, $DPOT_{L_0}$ maintains a stable ASR of around 15% across all training rounds, exceeding the random guess accuracy of 10% for the 10-class dataset. In the continuous attack pattern, $DPOT_{L_0}$ achieves a significant ASR, peaking at 80.03% during training and stabilizing around 40%, which is higher than the single-shot pattern. These results indicate that Flip is vulnerable to optimized triggers with varying appearances across different rounds, because recovering from backdoors is an after-effect strategy which is unable to stop new and distinct backdoors from injecting into the model.

Figure 7 illustrates the global model's performance on the main task data when using Flip as a defense while under $DPOT_{L_0}$

attack. We observed that employing Flip reduces the global model's main-task performance compared to not using it. In our baseline experiment on Fashion MNIST, with the same data distribution and aggregation rule (FedAvg), the model achieved an 86.7% MA. However, Flip's global model achieved only 82.8% MA at its best by the end, even with pre-trained model initialization. Additionally, under continuous attack by the $DPOT_{L_0}$ trigger, the global model's MA further declined compared to the less frequent attack pattern. This raises concerns about Flip's ability to maintain stable and normal performance on the main-task while effectively defending against attacks.

In summary, Flip represents an early effort to explore client-side defenses that do not rely on analyzing clients' model updates. While it demonstrates better defense effectiveness against $DPOT_{L_0}$ compared to server-side defenses, concerns about its potential impact on main-task convergence warrant further investigation.

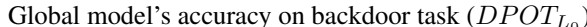

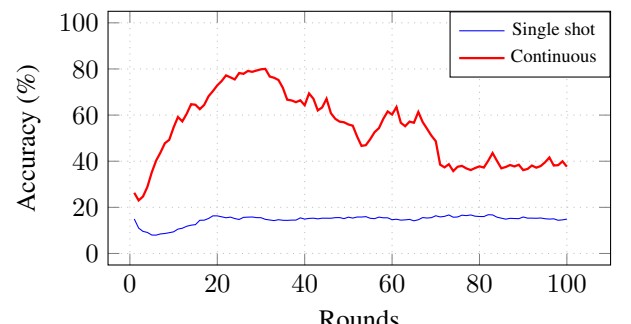
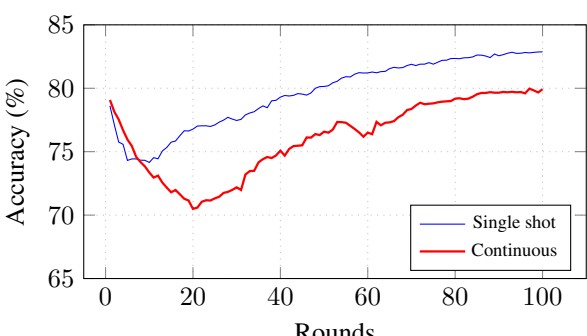

*Figure 6.* Global model's Attack Success Rate under $DPOT_{L_0}$ attack when employed Flip as defense strategy. (Fashion MNIST)

*Figure 7.* Global model's Main-task Accuracy under $DPOT_{L_0}$ attack when employed Flip as defense strategy. (Fashion MNIST)

## H. Evaluation of $DPOT_{L_0}$ attack against FRL (Mozaffari et al., 2023)

FRL (Mozaffari et al., 2023) is a defense strategy where the server sparsifies the value space of model updates, allowing clients to vote on the most effective model updates based on their local data. The server then aggregates only the accepted votes while rejecting outliers to construct the global model. In this section, we evaluate the effectiveness of the $DPOT_{L_0}$ attack against the FRL defense. Similar to the experiment on Flip, we implemented our attack on FRL's open-source project by injecting our data-poisoning algorithms into a portion of clients' execution and removing any inconsistent steps that distinguished malicious clients from benign ones during training. We used FRL's default settings, in which only 2% of clients were malicious, and tested our attack on the CIFAR10 dataset as the main training task.

Table 10 presents the performance results of the $DPOT_{L_0}$ attack on an FL system employing FRL as the defense method. The ASR of $DPOT_{L_0}$ (92.5%) is significantly higher than that of other backdoor attack approaches tested and discussed in FRL's paper. This indicates that FRL, which relies on analyzing clients' model updates, is vulnerable to our attack. The evaluation results also demonstrate that the $DPOT_{L_0}$ attack is more advanced than backdoor attacks with static objectives when targeting the FRL defense strategy.

*Table 10.* Comparison results on CIFAR10.

| Attacks | ASR |
|---|---|
| Semantic backdoor attacks | 49.2 |
| Artificial backdoor attacks | 0 |
| Edge-Case backdoor attacks | 64.6 |
| $DPOT_{L_0}$ **backdoor attacks** | **92.5** |

## I. Evaluation of $DPOT_{L_0}$ attack against BackdoorIndicator (Li & Dai, 2024)

We conducted experiments with different learning rates to demonstrate $DPOT_{L_0}$'s attack effectiveness against BackdoorIndicator, comparing it to Fixed pixel-pattern Triggers (FT).

*Table 11.* Comparison of $DPOT_{L_0}$ and FT's ASR against BackdoorIndicator.

| Learning Rate | 0.01 | 0.025 | 0.05 |
|---|---|---|---|
| Fixed pixel-pattern (Final ASR) | 10.7 | 23.3 | 26.3 |
| DPOT (Final ASR) | **100** | **99.9** | **99.9** |
| DPOT (Avg ASR) | **70.5** | **89.6** | **91.2** |

As shown in the table above, the $DPOT_{L_0}$ trigger maintains a significant Final ASR ($> 50\%$) against BackdoorIndicator across different learning rates and outperforms FT. We observe that BackdoorIndicator's defense effectiveness improves with smaller learning rates, consistent with the results in its original paper.

## J. Trigger size selection

We determined the size of the $DPOT_{L_0}$ trigger for each FL task by balancing its subtlety with achieving an effective ASR. A trigger's subtlety was evaluated by measuring the accuracy drop it caused when an un-attacked model predicted poisoned images into their original benign labels.

An un-attacked model with the same architecture as the victim FL system's model was used to assess the accuracy drop. The results for different datasets are presented in Table 12.

*Table 12.* Impact of $DPOT_{L_0}$ trigger size on un-attacked models' accuracy

| | **Trigger size** | **0** | **25** | **64** | **100** |
|---|---|---|---|---|---|
| Fashion-MNIST | Clean label acc | 85.76 | 79.32 | 76.07 | 70.53 |
| | Drop (%) | 0 | 7.5 | **11.30** | 17.76 |
| FEMNIST | Clean label acc | 81.24 | 68.11 | 45.12 | 28.39 |
| | Drop (%) | 0 | **16.16** | 44.46 | 65.05 |
| CIFAR10 | Clean label acc | 70.81 | 52.98 | 35.90 | 25.06 |
| | Drop (%) | 0 | **25.18** | 49.30 | 64.61 |
| Tiny-ImageNet | Clean label acc | 43.44 | 42.32 | 35.89 | 29.53 |
| | Drop (%) | 0 | 2.58 | **17.38** | 32.02 |

- Benign acc: accuracy of poisoned data being predicted to its original bengin label.

- Drop (%): Benign acc drop compared to when testing clean data (trigger size is 0) on the same un-attacked model.

We established a 30% upper limit for the acceptable accuracy drop and a minimum final ASR effectiveness threshold of 50%. The smallest trigger size meeting both criteria was chosen. Notably, Table 12 reveals that the sensitivity of accuracy drop to trigger sizes varies across datasets and model architectures.

## K. Visualization of Triggers

### K.1. FT, DFT, and $DPOT_{L_0}$ triggers on Tiny ImageNet images

We displayed FT, DFT, and $DPOT_{L_0}$ triggers on images from the Tiny ImageNet dataset in Figures 8, 9, and 10.

### K.2. $DPOT_{L_0}$ triggers on images from different datasets.

We displayed $DPOT_{L_0}$ triggers generated for different datasets in Figure 11.

### K.3. Trigger evolution during training

In Figure 14 and Figure 15, we demonstrated how $DPOT_{L_0}$ trigger changes during the FL training.

In Figure 14, we showed one screenshot of the trigger on a blank background in the same size of the cifar10's figure for every ten global rounds. These trigger screenshots were collected during a $DPOT_{L_0}$ attacking experiment that trains ResNet18 as the global model on the CIFAR-10 dataset, with Trimmed Mean used as the aggregation rule. Figure 12 displays the MA and ASR of the global model over 150 global rounds in this experiment.

Similarly, in Figure 15 we showed one screenshot of the trigger on a blank background in the same size of the Tiny

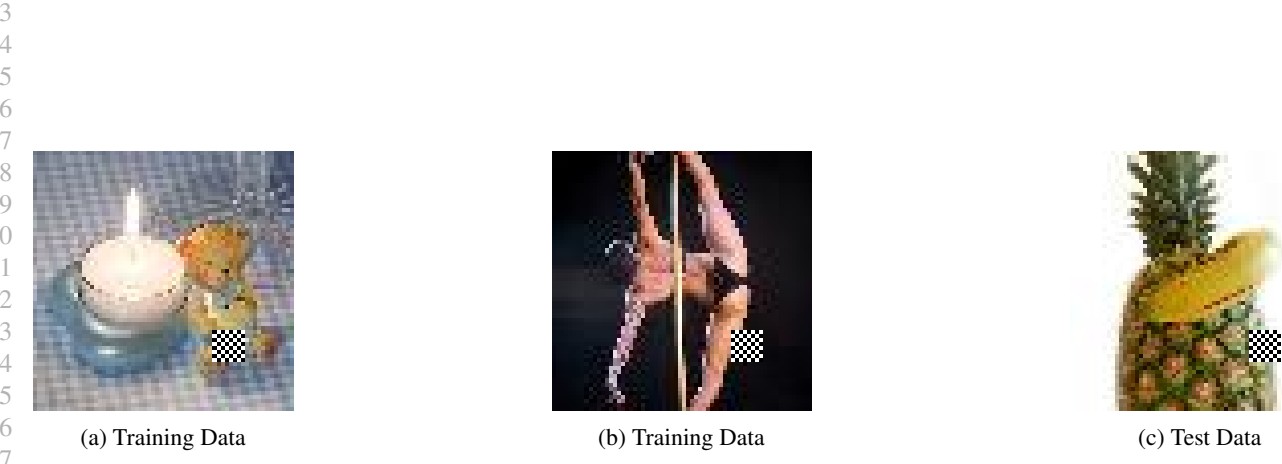

(a) Training Data        (b) Training Data        (c) Test Data

*Figure 8.* FT trigger on Tiny ImageNet data. Training Data 8a and 8b are from different malicious clients. Test Data 8c is used to test ASR.

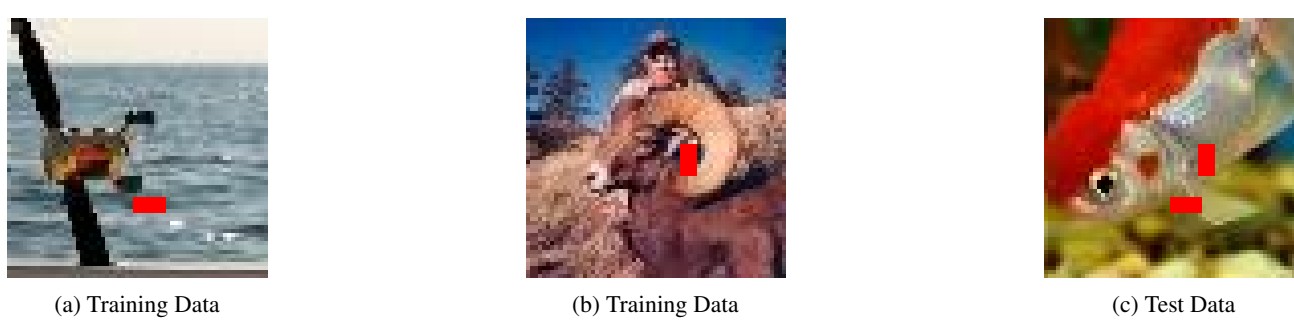

(a) Training Data        (b) Training Data        (c) Test Data

*Figure 9.* DFT trigger on Tiny ImageNet data. Training Data 9a and 9b are from different malicious clients. Test Data 9c is used to test ASR.

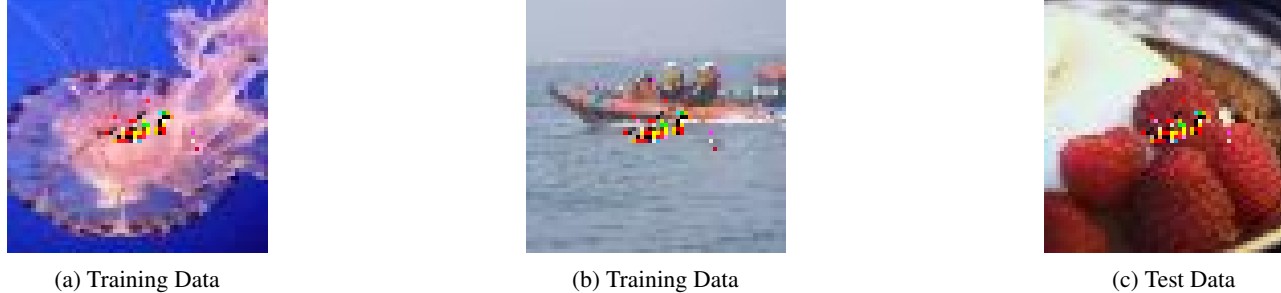

(a) Training Data        (b) Training Data        (c) Test Data

*Figure 10.* $DPOT_{L_0}$ trigger on Tiny ImageNet data. Training Data 10a and 10b are from different malicious clients. Test Data 10c is used to test ASR.

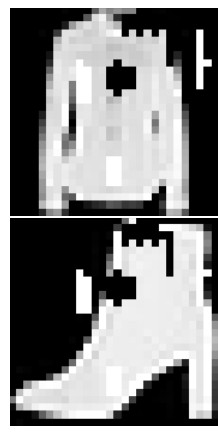 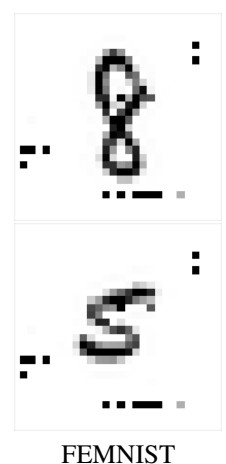 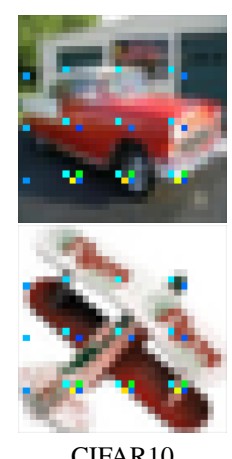 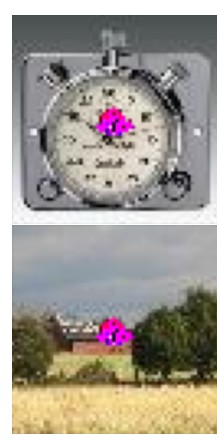

| Fashion MNIST | FEMNIST | CIFAR10 | Tiny ImageNet |

*Figure 11.* $DPOT_{L_0}$ triggers on images from different datasets.

ImageNet's figure for every ten global rounds. These trigger screenshots were collected during a $DPOT_{L_0}$ attacking experiment that trains VGG11 as the global model on the Tiny ImageNet dataset, with Trimmed Mean used as the aggregation rule. Figure 13 displays the MA and ASR of the global model over 100 global rounds in this experiment.

Figures 14 and 15 show that the $DPOT_{L_0}$ trigger evolves gradually and coherently over rounds, reflecting its dependency on the global model- the trigger is optimized based on the global model, and the global model is influenced by malicious updates tied to the trigger. As the global model evolves, the $DPOT_{L_0}$ trigger follows a similar consistent pattern.

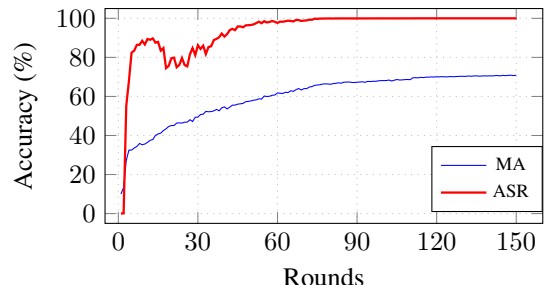

*Figure 12.* Global model's accuracy in experiment of getting trigger screenshots in Figure 14. (CIFAR10, ResNet18)

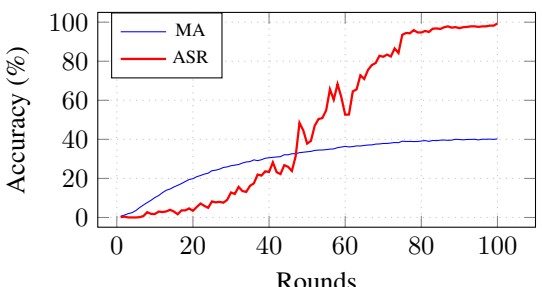

*Figure 13.* Global model's accuracy in experiment of getting trigger screenshots in Figure 15. (Tiny ImageNet, VGG11)

## L. Impact of Malicious Client Ratio (MCR)

In this section, we evaluated the impact of different Malicious Client Ratios (MCR) on the attacking performance of $DPOT_{L_0}$ attack. We assumed that the number of malicious clients in the FL system should be kept small ($\leq 30\%$) for practical reasons. We varied the MCR across four different settings (0.05, 0.1, 0.2, and 0.3) while keeping other settings consistent with those in Table 7. We experimented over 10 different defenses on the learning tasks of the CIFAR10 datasets and compare $DPOT_{L_0}$'s results with FT and DFT.

Tables 13 presents the evaluation results of attack effectiveness. $DPOT_{L_0}$ exhibited a dominant advantage over FT and DFT when the MCR is small (0.05 and 0.1). However, this advantage diminished with increasing MCR, indicating that when a sufficient number of malicious clients present in FL, even FT and DFT can achieve respectable $ASR$ against certain defense strategies. In most cases, the $ASR$ for all attacks continued to rise as the MCR increased, with the exception of FLAME. Results obtained with FLAME indicate that the number of malicious clients did not significantly impact its defense effectiveness. Table 14 presents the Main-task Accuracy results for each experiment considered in this section.

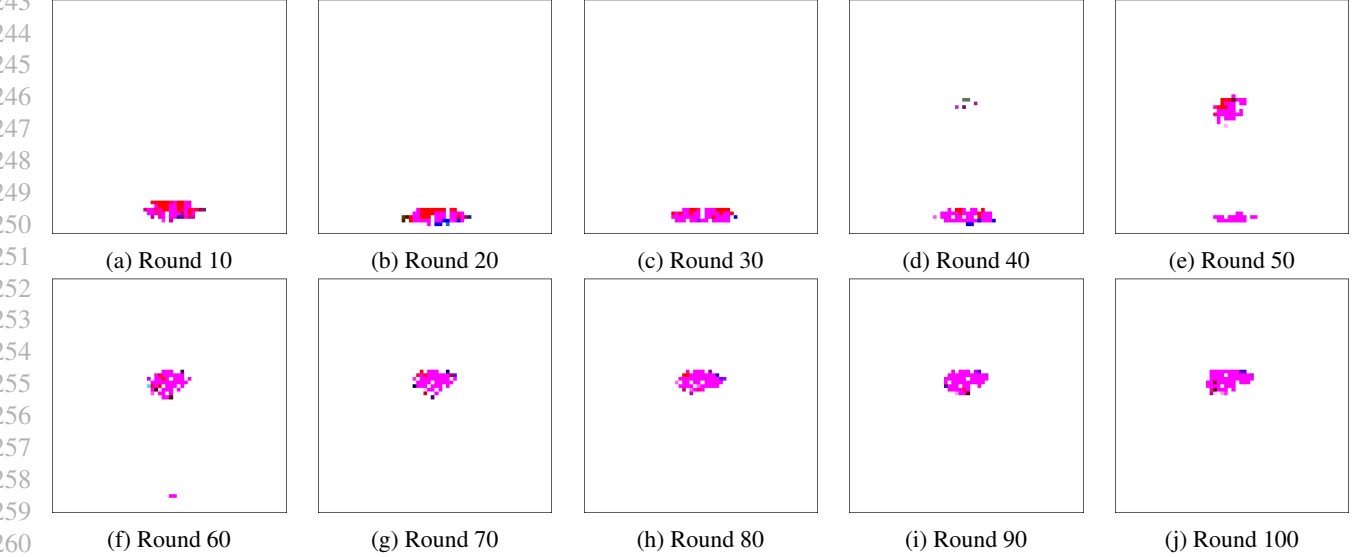

*Figure 14.* (CIFAR10, ResNet18) $DPOT_{L_0}$ triggers on different rounds.

*Figure 15.* (Tiny ImageNet, VGG11) $DPOT_{L_0}$ triggers on different rounds.

*Table 13.* The attack effectiveness under different MCR(CIFAR10).

| MCR | Final $ASR$ | | | | | | | | | | | | Average $ASR$ | | | | | | | | | | | |
|---|---|---|---|---|---|---|---|---|---|---|---|---|---|---|---|---|---|---|---|---|---|---|---|---|
| | 0.05 | | | 0.1 | | | 0.2 | | | 0.3 | | | 0.05 | | | 0.1 | | | 0.2 | | | 0.3 | | |
| | Ours | FT | DFT | Ours | FT | DFT | Ours | FT | DFT | Ours | FT | DFT | Ours | FT | DFT | Ours | FT | DFT | Ours | FT | DFT | Ours | FT | DFT |
| FedAvg | 100 | 100 | 93 | 100 | 100 | 100 | 100 | 100 | 100 | 100 | 100 | 100 | 99 | 88 | 50 | 99 | 96 | 88 | 99 | 99 | 92 | 99 | 100 | 97 |
| Median | 100 | 81 | 72 | 100 | 100 | 97 | 100 | 100 | 100 | 100 | 100 | 100 | 96 | 47 | 42 | 97 | 79 | 63 | 99 | 97 | 82 | 99 | 98 | 93 |
| Trimmed Mean | 100 | 95 | 38 | 100 | 100 | 99 | 100 | 100 | 100 | 100 | 100 | 100 | 89 | 59 | 23 | 98 | 82 | 69 | 99 | 94 | 85 | 99 | 99 | 92 |
| RobustLR | 100 | 100 | 100 | 100 | 100 | 100 | 100 | 100 | 100 | 100 | 100 | 100 | 99 | 94 | 87 | 99 | 98 | 94 | 99 | 99 | 98 | 99 | 99 | 99 |
| RFA | 100 | 100 | 98 | 100 | 100 | 100 | 100 | 100 | 100 | 100 | 100 | 100 | 98 | 81 | 55 | 99 | 95 | 90 | 99 | 99 | 97 | 99 | 99 | 98 |
| FLAIR | 62 | 15 | 10 | 58 | 25 | 9 | 67 | 27 | 22 | 82 | 33 | 40 | 51 | 14 | 10 | 64 | 24 | 9 | 68 | 24 | 16 | 84 | 42 | 30 |
| FLCert | 99 | 93 | 34 | 100 | 100 | 95 | 100 | 100 | 100 | 100 | 100 | 100 | 88 | 60 | 21 | 98 | 87 | 60 | 98 | 94 | 83 | 99 | 99 | 91 |
| FLAME | 60 | 19 | 17 | 52 | 18 | 51 | 50 | 16 | 16 | 55 | 19 | 16 | 56 | 18 | 14 | 66 | 19 | 34 | 53 | 19 | 16 | 70 | 23 | 43 |
| FoolsGold | 100 | 100 | 94 | 100 | 100 | 100 | 100 | 100 | 100 | 100 | 100 | 100 | 98 | 88 | 53 | 99 | 97 | 87 | 99 | 99 | 95 | 99 | 99 | 98 |
| Multi-Krum | 100 | 100 | 100 | 100 | 100 | 100 | 100 | 100 | 100 | 100 | 100 | 100 | 99 | 99 | 83 | 99 | 100 | 98 | 98 | 100 | 99 | 99 | 100 | 100 |

*Table 14.* The Main-task Accuracy (MA) under different MCR (CIFAR10).

| MCR | | 0.05 | | | 0.1 | | | 0.2 | | | 0.3 | | |
|---|---|---|---|---|---|---|---|---|---|---|---|---|---|
| | None | Ours | FT | DFT | Ours | FT | DFT | Ours | FT | DFT | Ours | FT | DFT |
| FedAvg | 70.3 | 70.66 | 70.37 | 71.37 | 70.03 | 71.04 | 70.13 | 69.9 | 70.39 | 71.18 | 70.25 | 70.69 | 70.24 |
| Median | 70.21 | 69.06 | 69.76 | 69.71 | 69.32 | 69.17 | 70.12 | 68.23 | 69.05 | 68.87 | 68.49 | 68.47 | 67.82 |
| Trimmed Mean | 69.43 | 70.42 | 70.24 | 70.84 | 69.9 | 69.17 | 69.78 | 69.33 | 69.19 | 69.8 | 69.23 | 68.83 | 68.02 |
| RobustLR | 70.35 | 70.10 | 70.35 | 70.48 | 70.58 | 70.42 | 69.90 | 70.31 | 70.56 | 70.43 | 70.05 | 69.11 | 69.22 |
| RFA | 70.42 | 70.69 | 70.27 | 70.77 | 70.35 | 70.44 | 70.16 | 70.72 | 70.33 | 69.56 | 70.09 | 69.72 | 69.37 |
| FLAIR | 70.25 | 70.62 | 71.04 | 70.42 | 69.80 | 71.45 | 70.89 | 71.85 | 71.20 | 71.16 | 71.26 | 69.74 | 70.99 |
| FLCert | 69.6 | 69.95 | 69.76 | 70.42 | 69.44 | 69.44 | 69.45 | 69.28 | 69.25 | 69.73 | 68.54 | 69.06 | 68.24 |
| FLAME | 70.14 | 70.28 | 70.93 | 70.85 | 69.62 | 70.87 | 71.01 | 70.71 | 70.4 | 70.58 | 69.19 | 71.45 | 70.52 |
| FoolsGold | 70.42 | 71.02 | 71.19 | 71.68 | 70.71 | 71.32 | 71.27 | 70.45 | 70.38 | 70.82 | 70.12 | 69.97 | 69.97 |
| Multi-Krum | 61.38 | 62.98 | 63.16 | 60.80 | 61.44 | 62.89 | 62.09 | 59.38 | 61.26 | 63.70 | 60.28 | 64.02 | 62.96 |

All $MA$ results for different attacks remain similar to the baseline $MA$, indicating the main-task convergence capability of pixel-pattern triggers.

# M. Impact of Trigger Size

Trigger Size, determining how many pixels in an image we can alter, is an important parameter for $L_0$-norm-bounded triggers. In this section, we assessed the impact of different trigger sizes on the performance of different $L_0$-norm-bounded triggers. We explored trigger sizes across four different settings (9, 25, 49, and 100) while maintaining other settings in accordance with those outlined in Table 7.

Tables 15 shows that $DPOT_{L_0}$ maintained a significant advantage in $ASR$ over FT and DFT across various trigger sizes, ranging from small to large. According to the results, we found that FT and DFT did not benefit from larger trigger sizes in achieving higher $ASR$ when encountering with some defenses such as FLAIR and FLAME. A possible explanation is that FT or DFT triggers in larger size can not cause smaller divergence between malicious and benign model updates, leaving malicious ones still susceptible to detection and filtering by defense mechanisms. In contrast, $DPOT_{L_0}$ demonstrated a continuous improvement in $ASR$ as the trigger size increased. Table 16 presents the Main-task Accuracy results for each experiment considered in this section. Results in it indicate all backdoor attacks achieved their main-task convergence goals during attacking.

*Table 15.* The attack effectiveness under different tirgger sizes (CIFAR10).

| Trigger Size | Final $ASR$ | | | | | | | | | | | | Average $ASR$ | | | | | | | | | | | |
|---|---|---|---|---|---|---|---|---|---|---|---|---|---|---|---|---|---|---|---|---|---|---|---|---|
| | 9 | | | 25 | | | 49 | | | 100 | | | 9 | | | 25 | | | 49 | | | 100 | | |
| | Ours | FT | DFT | Ours | FT | DFT | Ours | FT | DFT | Ours | FT | DFT | Ours | FT | DFT | Ours | FT | DFT | Ours | FT | DFT | Ours | FT | DFT |
| FedAvg | 100 | 94 | 49 | 100 | 100 | 93 | 100 | 100 | 91 | 100 | 100 | 77 | 95 | 60 | 28 | 99 | 88 | 50 | 99 | 90 | 59 | 99 | 93 | 52 |
| Median | 97 | 23 | 12 | 100 | 81 | 72 | 100 | 95 | 25 | 100 | 99 | 46 | 66 | 21 | 12 | 96 | 47 | 42 | 98 | 66 | 17 | 99 | 82 | 29 |
| Trimmed Mean | 98 | 51 | 14 | 100 | 95 | 38 | 100 | 99 | 43 | 100 | 100 | 74 | 71 | 29 | 13 | 89 | 59 | 23 | 99 | 74 | 27 | 99 | 79 | 44 |
| RobustLR | 100 | 100 | 100 | 100 | 100 | 100 | 100 | 100 | 98 | 100 | 100 | 99 | 95 | 91 | 69 | 99 | 94 | 87 | 99 | 94 | 77 | 99 | 95 | 82 |
| RFA | 100 | 100 | 99 | 100 | 100 | 98 | 100 | 100 | 100 | 100 | 100 | 98 | 93 | 79 | 56 | 98 | 81 | 55 | 99 | 81 | 71 | 99 | 90 | 73 |
| FLAIR | 27 | 14 | 14 | 62 | 15 | 10 | 89 | 22 | 15 | 99 | 24 | 14 | 24 | 14 | 13 | 51 | 14 | 10 | 84 | 22 | 15 | 98 | 16 | 13 |
| FLCert | 99 | 38 | 14 | 99 | 93 | 34 | 100 | 88 | 51 | 100 | 100 | 49 | 78 | 26 | 13 | 88 | 60 | 21 | 99 | 59 | 23 | 99 | 78 | 33 |
| FLAME | 21 | 18 | 12 | 60 | 19 | 17 | 100 | 12 | 11 | 100 | 33 | 31 | 35 | 17 | 12 | 56 | 18 | 14 | 84 | 17 | 11 | 90 | 31 | 24 |
| FoolsGold | 100 | 100 | 43 | 100 | 100 | 94 | 100 | 100 | 98 | 100 | 100 | 81 | 93 | 72 | 23 | 98 | 88 | 53 | 99 | 94 | 69 | 99 | 94 | 55 |
| Multi-Krum | 100 | 100 | 15 | 100 | 100 | 100 | 99 | 100 | 100 | 100 | 100 | 100 | 99 | 99 | 11 | 99 | 99 | 83 | 99 | 99 | 95 | 99 | 99 | 97 |

*Table 16.* The Main-task Accuracy (MA) under different trigger sizes. (CIFAR10).

| Trigger Size | | 9 | | | 25 | | | 49 | | | 100 | | |
|---|---|---|---|---|---|---|---|---|---|---|---|---|---|
| | None | Ours | FT | DFT | Ours | FT | DFT | Ours | FT | DFT | Ours | FT | DFT |
| FedAvg | 70.3 | 70.88 | 70.72 | 71.25 | 70.66 | 70.37 | 71.37 | 70.77 | 71.35 | 70.94 | 69.92 | 70.71 | 71.15 |
| Median | 70.21 | 68.31 | 70.04 | 68.69 | 69.06 | 69.76 | 69.71 | 69.95 | 70.54 | 70.56 | 69.88 | 70.30 | 70.86 |
| Trimmed Mean | 69.43 | 69.75 | 70.13 | 70.19 | 70.42 | 70.24 | 70.84 | 69.42 | 70.17 | 69.79 | 69.67 | 70.26 | 70.68 |
| RobustLR | 70.35 | 70.48 | 70.95 | 69.48 | 70.10 | 70.35 | 70.48 | 70.79 | 70.08 | 70.27 | 70.39 | 69.73 | 69.86 |
| RFA | 70.42 | 70.45 | 70.16 | 71.00 | 70.69 | 70.27 | 70.77 | 70.56 | 70.19 | 70.62 | 70.52 | 69.22 | 70.77 |
| FLAIR | 70.25 | 70.79 | 70.67 | 70.58 | 70.62 | 71.04 | 70.42 | 70.84 | 69.96 | 71.03 | 71.17 | 70.65 | 70.28 |
| FLCert | 69.6 | 69.88 | 69.64 | 69.87 | 69.95 | 69.76 | 70.42 | 67.77 | 69.83 | 70.08 | 68.81 | 70.81 | 70.41 |
| FLAME | 70.14 | 70.07 | 71.24 | 70.19 | 70.28 | 70.93 | 70.85 | 69.87 | 71.20 | 70.68 | 67.24 | 71.06 | 70.75 |
| FoolsGold | 70.42 | 70.4 | 72.1 | 70.09 | 71.02 | 71.19 | 71.68 | 70.66 | 70.75 | 71.38 | 69.84 | 71.06 | 71.64 |
| Multi-Krum | 61.38 | 62.86 | 64.65 | 58.90 | 62.98 | 63.16 | 60.80 | 58.23 | 60.16 | 64.04 | 63.03 | 61.64 | 63.33 |

# N. Impact of Data Poison Rate (DPR)

*Table 17.* Final ASR under different DPR.

| Data Poison Rate | 0.3 | | | 0.5 | | | 0.8 | | |
|---|---|---|---|---|---|---|---|---|---|
| | Ours | FT | DFT | Ours | FT | DFT | Ours | FT | DFT |
| Fedavg | **100.0** | **100.0** | 99.0 | **100.0** | **100.0** | 92.5 | **100.0** | **100.0** | 99.8 |
| Median | **100.0** | 97.0 | 61.0 | **100.0** | 81.3 | 72.0 | **100.0** | 96.3 | 35.9 |
| Trimmed Mean | **99.7** | 97.3 | 67.6 | **100.0** | 94.8 | 38.3 | **99.9** | 95.5 | 45.8 |
| RobustLR | **100.0** | **100.0** | 99.5 | **100.0** | **100.0** | 100.0 | **100.0** | **100.0** | 99.7 |
| RFA | 98.9 | **100.0** | 94.0 | **100.0** | **100.0** | 98.2 | **100.0** | **100.0** | 79.3 |
| FLAIR | **54.9** | 17.6 | 20.2 | **62.3** | 15.1 | 9.7 | **40.1** | 14.2 | 14.9 |
| FLCert | **99.4** | 96.7 | 63.5 | **99.2** | 92.6 | 33.7 | **99.9** | 89.1 | 34.8 |
| FLAME | **99.9** | 26.6 | 15.4 | **59.8** | 19.0 | 16.8 | **24.3** | 12.1 | 13.2 |
| FoolsGold | **100.0** | **100.0** | 98.8 | **100.0** | **100.0** | 94.1 | **100.0** | **100.0** | 98.3 |
| Multi-Krum | 95.2 | **100.0** | 77.3 | **100.0** | **100.0** | 99.9 | **100.0** | **100.0** | **100.0** |

*Table 18.* Average ASR under different DPR.

| Data Poison Rate | 0.3 | | | 0.5 | | | 0.8 | | |
|---|---|---|---|---|---|---|---|---|---|
| | Ours | FT | DFT | Ours | FT | DFT | Ours | FT | DFT |
| Fedavg | **97.3** | 92.5 | 75.5 | **98.5** | 88.1 | 50.2 | **98.3** | 93.1 | 81.2 |
| Median | **93.4** | 68.2 | 34.2 | **96.1** | 46.6 | 41.8 | **96.4** | 59.2 | 22.7 |
| Trimmed Mean | **88.0** | 68.5 | 34.2 | **88.6** | 59.4 | 22.6 | **95.2** | 65.6 | 29.6 |
| RobustLR | **97.9** | 94.7 | 76.7 | **98.6** | 94.4 | 87.4 | **98.2** | 92.2 | 78.3 |
| RFA | 87.0 | 82.9 | 53.5 | **97.8** | 81.0 | 55.3 | **97.9** | 81.5 | 45.7 |
| FLAIR | **50.1** | 16.2 | 15.4 | **50.7** | 13.6 | 10.1 | **45.3** | 14.1 | 13.8 |
| FLCert | **93.8** | 65.7 | 33.1 | **88.3** | 59.5 | 21.3 | **91.4** | 61.8 | 23.4 |
| FLAME | **84.3** | 51.6 | 28.4 | **56.0** | 18.2 | 14.4 | 34.5 | **43.1** | 40.4 |
| FoolsGold | **97.9** | 91.6 | 72.8 | **98.5** | 88.4 | 53.0 | **97.7** | 89.6 | 71.9 |
| Multi-Krum | 92.8 | **98.3** | 56.1 | **98.7** | **98.7** | 82.8 | **99.6** | 98.7 | 99.1 |

*Table 19.* Main-task Accuracy under different DPR.

| Data Poison Rate | 0.3 | | | 0.5 | | | 0.8 | | |
|---|---|---|---|---|---|---|---|---|---|
| | Ours | FT | DFT | Ours | FT | DFT | Ours | FT | DFT |
| Fedavg | 71.2 | 70.3 | 70.2 | 70.7 | 70.4 | 71.4 | 69.3 | 70.3 | 70.1 |
| Median | 69.3 | 69.8 | 70.0 | 69.1 | 69.8 | 69.7 | 69.5 | 69.0 | 69.5 |
| Trimmed Mean | 69.9 | 69.8 | 69.3 | 70.4 | 70.2 | 70.8 | 68.6 | 68.9 | 69.6 |
| RobustLR | 70.4 | 70.3 | 70.7 | 70.4 | 70.4 | 70.4 | 70.0 | 70.7 | 71.1 |
| RFA | 70.4 | 70.4 | 70.5 | 70.7 | 70.7 | 70.0 | 70.6 | 70.5 | 70.0 |
| FLAIR | 69.9 | 69.2 | 69.2 | 70.6 | 71.0 | 70.4 | 69.8 | 68.5 | 69.7 |
| FLCert | 70.7 | 70.1 | 68.9 | 70.0 | 69.8 | 70.4 | 69.1 | 67.9 | 69.9 |
| FLAME | 69.3 | 70.5 | 70.7 | 70.3 | 70.9 | 70.9 | 68.6 | 68.5 | 69.3 |
| FoolsGold | 70.5 | 70.7 | 71.4 | 71.0 | 71.2 | 71.7 | 70.1 | 70.0 | 70.4 |
| Multi-Krum | 62.8 | 60.1 | 59.0 | 62.9 | 63.2 | 60.8 | 60.0 | 61.1 | 61.1 |

We implemented experiments to study the impact of different DPR on the performance of DPOT, FT, and DFT attacks. We evaluated Final ASR and Avg ASR to compare attack effectiveness, and Main-task Accuracy to assess the main-task convergence. Other attack and FL training settings are consistent with those in Table 7. The results are shown in the Tables 17, 18, and 19.

In general, $DPOT_{L_0}$ shows better attack effectiveness than FT and DFT across different DPR values. An interesting observation is that the best attack effectiveness occurs at different DPR values for various defenses and attacks. One possible

explanation for this is that a smaller DPR weakens the impact of poisoned data on model updates, reducing the divergence between malicious and benign updates, which helps bypass defenses. On the other hand, a larger DPR increases the impact on model updates, speeding up the attack's effectiveness but also increasing the divergence, making malicious updates more detectable and filterable by defenses. Therefore, the attack effectiveness of different attacks against different defenses depends on how stealthy the attack can make the model updates at that DPR and how effectively the defense can mitigate those malicious updates at the same DPR.

## O. Impact of Non-iid degree

We implemented experiments to study the impact of different Non-iid degrees on the performance of $DPOT_{L_0}$, FT, and DFT attacks. We evaluated Final ASR and Avg ASR to compare attack effectiveness, and Main-task Accuracy to assess main-task convergence. Other attack and FL training settings are consistent with those in Table 7. The results are shown in Tables 20, 21, and 22.

It can be observed from the last table that different Non-iid degrees result in different main-task accuracies. A smaller Non-iid degree indicates that the data distribution is closer to an iid distribution, with a Non-iid degree of 0 representing an exact iid distribution. The $DPOT_{L_0}$ attack generally exhibits better attack effectiveness than FT and DFT across different Non-iid degree settings.

*Table 20.* Final ASR results in different Non-iid degrees.

| Non-iid degree | 0 | | | 0.2 | | | 0.5 | | | 0.8 | | |
|---|---|---|---|---|---|---|---|---|---|---|---|---|
| | Ours | FT | DFT | Ours | FT | DFT | Ours | FT | DFT | Ours | FT | DFT |
| Fedavg | **100.0** | **100.0** | 99.7 | **100.0** | **100.0** | 99.8 | **100.0** | **100.0** | 92.5 | **100.0** | **100.0** | 99.3 |
| Median | **100.0** | 99.4 | 72.2 | **100.0** | 99.0 | 66.8 | **100.0** | 81.3 | 72.0 | **99.3** | 99.2 | 33.0 |
| Trimmed Mean | **100.0** | 99.4 | 73.6 | **100.0** | 99.8 | 86.8 | **100.0** | 94.8 | 38.3 | **100.0** | 96.0 | 65.8 |
| RobustLR | **100.0** | **100.0** | 99.8 | **100.0** | **100.0** | 99.8 | **100.0** | **100.0** | 100.0 | **100.0** | **100.0** | 99.2 |
| RFA | **100.0** | **100.0** | 97.0 | **100.0** | 99.8 | 95.5 | **100.0** | **100.0** | 98.2 | **100.0** | **100.0** | 98.5 |
| FLAIR | **49.7** | 21.2 | 12.9 | **65.4** | 29.3 | 15.8 | **62.3** | 15.1 | 9.7 | **19.9** | 12.6 | 5.1 |
| FLCert | **99.9** | 99.7 | 77.1 | **100.0** | 96.7 | 54.1 | **99.2** | 92.6 | 33.7 | **100.0** | 92.1 | 19.4 |
| FLAME | **46.5** | 15.7 | 15.1 | **57.6** | 16.7 | 16.2 | **59.8** | 19.0 | 16.8 | **100.0** | 8.5 | 26.5 |
| FoolsGold | **100.0** | **100.0** | 99.9 | **100.0** | **100.0** | 99.6 | **100.0** | **100.0** | 94.1 | **100.0** | **100.0** | 99.9 |
| Multi-Krum | **100.0** | **100.0** | 95.2 | **100.0** | **100.0** | 18.9 | **100.0** | **100.0** | 99.9 | **100.0** | **100.0** | 100.0 |

*Table 21.* Average ASR results in different Non-iid degrees.

| Non-iid degree | 0 | | | 0.2 | | | 0.5 | | | 0.8 | | |
|---|---|---|---|---|---|---|---|---|---|---|---|---|
| | Ours | FT | DFT | Ours | FT | DFT | Ours | FT | DFT | Ours | FT | DFT |
| Fedavg | **98.1** | 94.6 | 81.5 | **97.9** | 90.8 | 80.9 | **98.5** | 88.1 | 50.2 | **98.3** | 94.2 | 75.5 |
| Median | **95.9** | 73.0 | 39.2 | **95.3** | 79.5 | 38.1 | **96.1** | 46.6 | 41.8 | **89.0** | 59.6 | 19.6 |
| Trimmed Mean | **95.3** | 75.6 | 42.6 | **96.7** | 80.5 | 53.7 | **88.6** | 59.4 | 22.6 | **93.3** | 49.4 | 33.6 |
| RobustLR | **98.5** | 95.0 | 79.5 | **98.6** | 92.9 | 81.7 | **98.6** | 94.4 | 87.4 | **98.1** | 95.1 | 71.0 |
| RFA | **95.8** | 86.6 | 61.4 | **97.5** | 87.9 | 61.5 | **97.8** | 81.0 | 55.3 | **98.0** | 85.8 | 58.6 |
| FLAIR | **56.7** | 27.7 | 12.1 | **64.9** | 28.2 | 14.5 | **50.7** | 13.6 | 10.1 | **21.0** | 9.3 | 5.4 |
| FLCert | **96.7** | 71.0 | 40.4 | **95.0** | 74.7 | 31.7 | **88.3** | 59.5 | 21.3 | **96.2** | 52.2 | 13.7 |
| FLAME | **63.0** | 38.8 | 26.0 | **56.1** | 28.3 | 21.4 | **56.0** | 18.2 | 14.4 | **73.8** | 21.1 | 37.8 |
| FoolsGold | **98.4** | 93.9 | 81.4 | **97.8** | 92.6 | 81.0 | **98.5** | 88.4 | 53.0 | **98.7** | 98.0 | 80.0 |
| Multi-Krum | 91.4 | **98.2** | 58.4 | **98.6** | 86.6 | 15.1 | **98.7** | **98.7** | 82.8 | 96.2 | **98.7** | 70.4 |

*Table 22.* Main-task Accuracy results in different Non-iid degrees.

| Non-iid degree | 0 | | | 0.2 | | | 0.5 | | | 0.8 | | |
|---|---|---|---|---|---|---|---|---|---|---|---|---|
| | Ours | FT | DFT | Ours | FT | DFT | Ours | FT | DFT | Ours | FT | DFT |
| Fedavg | 74.5 | 74.5 | 74.9 | 74.4 | 74.3 | 75.0 | 70.7 | 70.4 | 71.4 | 55.9 | 55.5 | 56.2 |
| Median | 74.5 | 75.5 | 74.3 | 73.8 | 74.3 | 74.9 | 69.1 | 69.8 | 69.7 | 53.4 | 52.9 | 54.5 |
| Trimmed Mean | 75.0 | 74.3 | 74.5 | 74.4 | 75.2 | 74.3 | 70.4 | 70.2 | 70.8 | 52.7 | 53.5 | 54.5 |
| RobustLR | 75.3 | 75.1 | 75.1 | 75.3 | 74.4 | 74.4 | 70.4 | 70.4 | 70.4 | 55.2 | 55.8 | 55.4 |
| RFA | 74.3 | 74.8 | 75.1 | 75.0 | 75.8 | 74.4 | 70.7 | 70.3 | 70.8 | 56.4 | 56.2 | 55.2 |
| FLAIR | 73.7 | 73.4 | 73.6 | 73.9 | 72.7 | 73.1 | 70.6 | 71.0 | 70.4 | 55.3 | 52.5 | 52.1 |
| FLCert | 74.6 | 74.0 | 74.7 | 74.1 | 74.8 | 74.8 | 70.0 | 69.8 | 70.4 | 53.9 | 53.5 | 53.3 |
| FLAME | 73.2 | 72.1 | 73.0 | 72.9 | 73.5 | 73.5 | 70.3 | 70.9 | 70.9 | 56.1 | 56.2 | 57.5 |
| FoolsGold | 74.1 | 74.5 | 74.5 | 74.6 | 74.3 | 74.5 | 71.0 | 71.2 | 71.7 | 57.0 | 55.1 | 55.1 |
| Multi-Krum | 73.8 | 73.6 | 73.1 | 73.4 | 73.2 | 73.0 | 62.9 | 63.2 | 60.8 | 38.7 | 41.0 | 38.4 |

## P. Different attacking patterns

In this section we evaluated $DPOT_{L_0}$ attack's extensive capability in attacking in different frequency and starting at different rounds. The results are shown in Table 23.

Table 23. $DPOT_{L_0}$'s attack effectiveness against FLAIR and FLAME across different start and interval rounds.

| Start round | Interval round | FLAIR | FLAME |
|---|---|---|---|
| 1 | 1 | 62.3 | **59.8** |
| 1 | 5 | 52.9 | 53.8 |
| 1 | 10 | 72.0 | 57.8 |
| 50 | 1 | 52.1 | 39.2 |
| 50 | 10 | **76.5** | 48.5 |
| 100 | 1 | 54.2 | 50.3 |
| 100 | 10 | 69.2 | 54.5 |

- Start round: The round that $DPOT_{L_0}$ starts attacking.

- Interval round: The number of rounds beween two adjacent $DPOT_{L_0}$ attacks.

In conclusion, $DPOT_{L_0}$ attack shows better attack effectiveness in lower attacking frequency when against specific defense strategy like FLAIR. FLAIR penalizes clients that are frequently flagged as suspicious by lowering their aggregation weight, reducing their impact on the global model. Malicious clients can regain their normal influence by pausing malicious behavior, allowing the penalty score to gradually decrease. For other defenses that do not use similar strategy (FLAME), the effectiveness of the $DPOT_{L_0}$ attack shows insignificant variation across different start rounds and interval rounds. A more successful attack effectiveness for FLAME however can be achieved by decreasing the Data Poison Rate as shown in Table 17.

## Q. Combine the scaling-based model poisoning techniques with $DPOT_{L_0}$

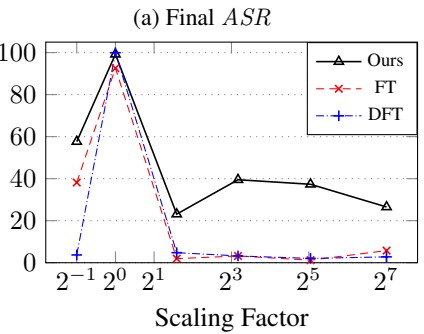
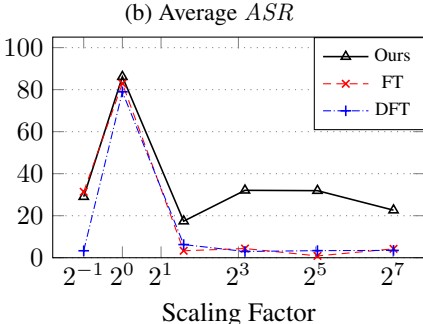

Figure 16. Comparison results of different attacks when employing the scaling-based model poisoning technique to undermine FLAME defense (implemented on the FEMNIST dataset).

In this section, we removed the TEEs assumption and conducted experiments to examine the effects of employing scaling-based model poisoning techniques on the attack performance of $DPOT_{L_0}$, FT, and DFT. By incorporating the model poisoning technique, our implementation of FT and DFT pipelines aligns more closely with the attack strategies introduced in state-of-the-art backdoor attacks on FL (Bagdasaryan et al., 2020; Xie et al., 2020).

Our experiments were designed within an FL system utilizing FLAME as its aggregation rule and FEMNIST dataset as its main training task. We adjusted the scaling factors, used to scale malicious clients' model updates, to be 0.5, 1, 3, 9, 33, and 129 respectively. Figures 16a and 16b illustrate the results of Final $ASR$ and Avg $ASR$ of various attacks in response to different scaling factors.

We observed that when the scaling factor is 1, all $DPOT_{L_0}$, FT, and DFT pipelines exhibit comparable and high $ASR$ against FLAME defense. However, as the scaling factor increases, FLAME demonstrates robust defense performance, significantly reducing the $ASR$ of every attack pipeline. Despite this mitigation, $DPOT_{L_0}$ shows greater resilience in

attack effectiveness compared to FT and DFT. The optimized trigger generated by our algorithms retains intrinsic attack effects on the global model even without successful data-poisoning techniques. When the scaling factor is reduced to 0.5, malicious model updates are expected to be stealthier, yet their contributions to the aggregated global model are also mitigated, resulting in reduced $ASR$ for all attack pipelines compared to when the scaling factor is 1.

