# OpenReview forum: "$DPOT_{L_0}$: Concealing Backdoored model updates in Federated Learning by Data Poisoning with $L_0$-norm-bounded Optimized Triggers"
_ICML.cc/2025/Conference — Submitted to ICML 2025_

### Official Review · Reviewer_1bCi · 2025-03-11

**Overall Recommendation:** 3

**Summary:**

The authors proposed a new backdoor attack method named DPOT. DPOT generates triggers by: 1) utilizing the sensitivity of the model's output concerning each pixel of the input to determine which positions should be selected, and 2) optimizing the pixel values of the triggers to maximize their effectiveness. The authors conducted numerous experiments to validate the performance of DPOT.

**Claims And Evidence:**

I am confused about the authors' threat model. I do not understand why a malicious client cannot manipulate their local training process (Lines 128-133).

**Essential References Not Discussed:**

Some federated backdoor attacks [1,2,3] also employ dynamic triggers.

[1] Iba: Towards irreversible backdoor attacks in federated learning

[2] Bad-PFL: Exploring Backdoor Attacks against Personalized Federated Learning

[3] Lurking in the shadows: Unveiling Stealthy Backdoor Attacks against Personalized Federated Learning

**Experimental Designs Or Analyses:**

Overall, the experiments are comprehensive. However, there are some issues. First, the authors do not specify the exact size of the triggers. Second, in federated learning, not every client is selected to participate in each round (Line 269).

**Methods And Evaluation Criteria:**

The method seems reasonable.

**Other Comments Or Suggestions:**

Since the authors used a Non-IID setting, they could test the proposed method's attack performance for personalized federated learning.

**Other Strengths And Weaknesses:**

The authors' method achieves state-of-the-art performance, and the experiments are very comprehensive.

The writing needs improvement. At present, the motivation behind the authors' method is limited.

**Questions For Authors:**

See above.

**Relation To Broader Scientific Literature:**

Although the authors present a new backdoor attack method, I currently do not see its practical value. In my view, it merely utilizes gradient information to determine the location of the triggers.

**Theoretical Claims:**

Since the triggers are added to the input data rather than the features, Proposition 5.1 cannot fully demonstrate the effectiveness of the authors' method.

---

> ### Author Rebuttal · Authors · 2025-03-28
>
> 1. I am confused about the authors' threat model. I do not understand why a malicious client cannot manipulate their local training process (Lines 128-133).
>
> We found this Wikipedia page very helpful for understanding our threat model: https://en.wikipedia.org/wiki/Trusted_execution_environment. It introduces the concept of TEEs and lists many commercial TEEs. Our threat model assumes that the local training process is executed within TEEs, which the FL server can verify by having each TEE attest to its software and inputs.  TEEs greatly increase the difficulty of malicious manipulation.
>
> 2. Since the triggers are added to the input data rather than the features, Proposition 5.1 cannot fully demonstrate the effectiveness of the authors' method.
>
> Following the problem setup of the feature learning theory paper (line 589), we consider a datum is composed by various patterns with some of patterns are critical for the classification, which we model as linear vectors and name them as features. Backdoor data's features are trigger patterns. We used $K$ to represent a dataset containing only benign features, and $K\cup K_{adv}$ to represent a dataset containing both benign and backdoor features. Proposition 5.1 correctly
> explains theoretical insights of our method in dataset-level, rather than data-level.
>
> 3. In federated learning, not every client is selected to participate in each round (Line 269).
>
> We experimented with scenarios where not every client is selected to participate in each round by randomly sampling a portion of clients for each round training, and used Selection Ratio (SR) to determine this portion. We present results for SR = 0.5 and SR = 1, comparing $DPOT_{L_0}$ with FT and DFT against three different defense strategies on CIFAR-10 as the main training task. As shown in the following table, $DPOT_{L_0}$ demonstrates better attack effectiveness in both SR settings and achieves a sufficient ASR faster than the other attacks.
>
> |   | | **Final**|  **ASR** | ----- | **Avg** | **ASR** |-----| **MA** | -----| -----| **Rounds to** | **achieve** |**ASR > 50** |
> |----|--------|---------------|--------|--------|---------------|--------|--------|----------------|--------|--------|-------|--------|--------|
> |   | | **Ours** | **FT**| **DFT**| **Ours**| **FT**| **DFT**| **Ours** | **FT**| **DFT**| **Ours** | **FT**| **DFT**|
> | **SR = 0.5** | Fedavg| **100**| **100.0**| 97| **98.6** | 90| 65| 70.5 | 70.3  | 69.8  | **2**| 16| 47|
> |- | Trimmed Mean   | **100**| **100.0**| 70.4  | **95**| 75.1  | 37.5  | 70.4 | 69.6  | 70.0  | **3**| 34| 104   |
> |    -  | FoolsGold| **100**| **100.0**| 97.5  | **98.8** | 93.9  | 73.6  | 70.2 | 70.3  | 69.6  | **2**| 7 | 33|
> | **SR = 1**   | Fedavg| **100**| **100.0**| 92.5  | **98.5** | 88.1  | 50.2  | 70.7 | 70.4  | 71.4  | **3**| 21| 80|
> |- | Trimmed Mean   | **100**| 94.8  | 38.3  | **88.6** | 59.4  | 22.6  | 70.4 | 70.2  | 70.8  | **7**| 65| - |
> |    -  | FoolsGold| **100**| **100.0**| 94.1  | **98.5** | 88.4  | 53| 71   | 71.2  | 71.7  | **3**| 18| 74|
>
>
>
> 4. Although the authors present a new backdoor attack method, I currently do not see its practical value. In my view, it merely utilizes gradient information to determine the location of the triggers.
>
> $DPOT_{L_0}$ considers a FL settings where all clients are run in a TEE environment, which has strict limitation over adversaries' capability in comparison to previous works. This highlights the enhanced practical value of $DPOT_{L_0}$.
>
> 5. Some federated backdoor attacks [1,2,3] also employ dynamic triggers.
>
> [1] Iba is one of the baselines that we did thorough comparison with (line 281, 311). Please see section 6.6 for our comparison results from 4 different dimensions. We will respectively mention reference [2], [3] for their contribution of employing dynamic triggers against Personalized Federated Learning in the related work, and thank you for sharing these references with us.
>
> 6. The writing needs improvement. At present, the motivation behind the authors' method is limited.
>
> We'd love to adopt suggestions to improve our writing. The motivation behind $DPOT_{L_0}$ is to expose the vulnerabilities in current FL defenses and emphasize the need for more robust countermeasures by demonstrating the effectiveness of the introduced data-poisoning-only backdoor attacks.
>
> 7. Since the authors used a Non-IID setting, they could test the proposed method's attack performance for personalized federated learning.
>
> Our work focuses on studying the security performance of the fundamental FL structure introduced by McMahan et al. (2017), with selected attack and defense baselines all served for this structure. We agree that exploring the attack performance of $DPOT_{L_0}$ on advanced FL structures, such as Personalized Federated Learning, Vertical Federated Learning, and Federated Transfer Learning, would be a highly promising direction for future work.

---

### Official Review · Reviewer_KkLM · 2025-03-13

**Overall Recommendation:** 3

**Summary:**

This paper introduces $DPOT_{L_{0}}$, a new backdoor attack method in FL that dynamically optimizes an L0-norm-bounded trigger to conceal malicious model updates among benign ones. By focusing on data poisoning alone, the attack avoids reliance on model poisoning, which is increasingly impractical under Trusted Execution Environments (TEEs). The authors theoretically justify the concealment property of DPOT_{L₀} in linear models and empirically demonstrate its effectiveness across four datasets and 12 defense strategies, outperforming existing methods in attack success rate (ASR) while maintaining main-task accuracy. The work highlights vulnerabilities in current FL defenses and underscores the need for more robust countermeasures.

**Claims And Evidence:**

Yes

**Essential References Not Discussed:**

N/A

**Experimental Designs Or Analyses:**

Yes.

**Methods And Evaluation Criteria:**

Yes

**Other Comments Or Suggestions:**

N/A

**Other Strengths And Weaknesses:**

Strengths:
* This paper considers a FL settings where all clients are run in a TEE environment, which has strick limitation over adversaries' capability in comparison to previous works.
* The evaluation is comprehensive, including a range of baseline attacks and benchmarks.
* The intuition and algorithms are clearly explained and easy to follow.

Weaknesses:
* What is the direct relationship between Section 5 and  Section 4? There is no clear explanation provided to demonstrate how the design of DPOT is benefited from the theoretical analysis. I would appreciate a brief explanation here to emphasize how Section 5 can help the reader  better understand DPOT.
* Section 4.2 is confusing. If the trigger size in the following part is simply pre-defined, it is unnecessary to have this section here. Besides, how the trigger size is chosen for the evaluation?

**Questions For Authors:**

N/A

**Relation To Broader Scientific Literature:**

This work proposes a new backdoor attack with a L0 regularization. The idea is a direct extension from previous optimized trigger backdoor attacks.

**Theoretical Claims:**

N/A

---

> ### Author Rebuttal · Authors · 2025-03-28
>
> 1. What is the direct relationship between Section 5 and Section 4? There is no clear explanation provided to demonstrate how the design of DPOT is benefited from the theoretical analysis. I would appreciate a brief explanation here to emphasize how Section 5 can help the reader better understand DPOT.
>
> Thank you for this valuable feedback and advice. Section 5 provided the theoretical insights that the difference in update directions between benign and malicious objectives is bounded by the error of the malicious data on the model. Section 4 introduced how $DPOT_{L_0}$ decreases backdoor data's error (loss) on the global model, and section 5 provided justification that the error reduction can help conceal malicious model updates among benign ones. We will add the following formula to paper for better connecting section 4 and section 5.
>
> $\min \epsilon_{adv} = \min_{\tau} \quad \frac{1}{|D|} \sum_{x \in D} \mathit{Loss}(W_g(x \odot \tau), y_t)$
>
> By optimizing $L_0$-norm bounded trigger $\tau$ to minimize the error $\epsilon_{adv}$, $DPOT_{L_0}$ not only reduces backdoor loss but also conceals malicious model updates (proposition 5.1).
>
> 2. Section 4.2 is confusing. If the trigger size in the following part is simply pre-defined, it is unnecessary to have this section here. Besides, how the trigger size is chosen for the evaluation?
>
> Thank you for this feedback. Section 4.2 introduces the strategy we use to pre-define the trigger size. We determined the trigger size by ensuring that the accuracy drop of poisoned data, predicted as benign by an un-attacked model, does not exceed 30% (subtlety goal). Concrete examples of trigger size selection used in evaluation are provided in Appendix J due to space limitations, as referenced in line 418. Based on your valuable feedback, we will improve the clarity of our statement and mention those examples earlier in the method section for better readability.
>
> *We found an experiment suggested by Reviewer Vb6j very interesting, and would like to share it here.*
>
> 3. Duration of attack effectiveness
>
> We compared $DPOT_{L_0}$ and $FT$ on the duration of their effectiveness after attack termination. Using FedAvg on the CIFAR-10 dataset, we first applied 50 rounds of data poisoning for both attacks reached 100% ASR. We then stopped the poisoning and recorded the training rounds needed for ASR to drop to 50%.
>
> Testing with different learning rates (lr), we found that higher lr shortened the attack's duration. As shown in the table, $DPOT_{L_0}$ consistently had a longer duration than $FT$ across all lr. A possible explanation is that $DPOT_{L_0}$ disperses trigger pixels across the image, activating more neurons and reinforcing trigger feature retention. In contrast, $FT$ clusters trigger pixels in a corner, affecting fewer neurons and leading to faster forgetting.
>
> |lr| 0.01 | 0.015 | 0.02 |
> |---|---|---|---|
> |$FT$|275|155|65|
> |$DPOT_{L_0}$|330|328|101|

---

### Official Review · Reviewer_gxWe · 2025-03-13

**Overall Recommendation:** 4

**Summary:**

This paper presents a method to optimize backdoor attack triggers in federated learning systems. The proposed scheme is validated by experiments. The main contributions of this work include: 1. Proposing a simple and effective method for generating triggers, simultaneously optimizing the pixel values and positions of the triggers. 2. Proposing a defense method based on statistical information to resist the proposed backdoor attack. This work conducts research from both attack and defense perspectives, making it a comprehensive study.

**Claims And Evidence:**

The claims made in the paper are reasonable and verifiable

**Essential References Not Discussed:**

N/A

**Experimental Designs Or Analyses:**

All the results of the experimental demonstration part of the paper are checked.

**Methods And Evaluation Criteria:**

The evaluation method used is reasonable

**Other Comments Or Suggestions:**

N/A

**Other Strengths And Weaknesses:**

This paper presents a method to optimize triggers in backdoor attacks. My main concerns are as follows:
1. As can be seen from Figure 2, the generated trigger is distinguishable by the human eye. Does this lead to smart clients using simple data filtering/clear methods to suppress backdoor attacks?
2. What insights does the theoretical analysis provided in section 5 specifically provide for the design of backdoor attacks?

**Questions For Authors:**

Simple defenses against poisoning data being filtered and deleted need to be discussed to reveal the actual value of the proposed methods

**Relation To Broader Scientific Literature:**

The content of this paper is related to the security and vulnerability of distributed training, and the proposed method is related to the robustness of distributed system

**Theoretical Claims:**

The theorem provided in this paper is reasonable

---

> ### Author Rebuttal · Authors · 2025-03-28
>
> 1. As can be seen from Figure 2, the generated trigger is distinguishable by the human eye. Does this lead to smart clients using simple data filtering/clear methods to suppress backdoor attacks? (Simple defenses against poisoning data being filtered and deleted need to be discussed to reveal the actual value of the proposed methods)
>
> We agree that our generated triggers are distinguishable by the human eyes.
>
> During the FL training, malicious clients as the attack executors should not want to filter triggers out, and benign clients without having trigger information are unable to filter triggers out. Even though we let TEE carry out clear methods on data, backdoor data still cannot transform to benign data since their labels have been changed. An adaptive change on attack can be constraining both $L_\infty$ and $L_0$ bounds of trigger during optimization so that the magnitude of trigger pixels can be better controlled.
>
> During the inference stage, smart users can clear the images before inputing them into the victim FL model to bypass backdoor attacks, but this clear method might also alter features in benign images which will degrade main-task accuracy. Nonetheless, we acknowledge that improving data cleansing methods to more accurately filter backdoor triggers is a valuable defense strategy worth exploring.
>
> 2. What insights does the theoretical analysis provided in section 5 specifically provide for the design of backdoor attacks?
>
> Thank you for this valuable feedback. Section 5 provided the theoretical insights that the difference in update directions between benign and malicious objectives is bounded by the error of the malicious data on the model. Section 4 introduced how $DPOT_{L_0}$ decreases backdoor data's error (loss) on the global model, and section 5 provided justification that the error reduction can help conceal malicious model updates among benign ones. We will add the following formula to paper for better connecting section 4 and section 5.
>
> $\min \epsilon_{adv} = \min_{\tau} \quad \frac{1}{|D|} \sum_{x \in D} \mathit{Loss}(W_g(x \odot \tau), y_t)$
>
> By optimizing $L_0$-norm bounded trigger $\tau$ to minimize the error $\epsilon_{adv}$, $DPOT_{L_0}$ not only reduces backdoor loss but also conceals malicious model updates (proposition 5.1).

---

### Official Review · Reviewer_Vb6j · 2025-03-21

**Overall Recommendation:** 3

**Summary:**

The paper proposes $DPOT_{L_0}$, a backdoor attack strategy for Federated Learning that focuses on concealing malicious model updates. Unlike traditional backdoor attacks that use fixed triggers or obvious model poisoning, $DPOT_{L_0}$ dynamically optimizes an $L_0$-norm-bounded trigger for each round. This trigger is designed to be subtle (minimally impacting data) and to align the malicious updates with benign updates, making detection difficult. The core idea is to create a per-round backdoor objective by optimizing a small number of pixels ($L_0$ constraint) to still achieve the backdoor goal (misclassification to a target label) but minimize the deviation from the expected model update direction. The paper includes theoretical justification for linear models and experimental evaluation on several datasets (FashionMNIST, FEMNIST, CIFAR10, Tiny ImageNet) and against multiple defenses.

**Claims And Evidence:**

- Claim 1: $DPOT_{L_0}$ effectively conceals malicious model updates.
    - Theoretical Justification (Proposition 5.1): Shows that in a linear model, the difference in update directions between benign and malicious objectives is bounded by the error of the malicious data on the model.
    - Experimental Results (Tables 1, 8, 9, and Figure 4): Demonstrate high attack success rates (ASR) across various datasets and defenses, often outperforming other attack methods.
    - Comparison to other optimized triggers (Section 6.6): Shows superior performance compared to L₂-norm-bounded triggers and partially optimized L₀-norm triggers, highlighting the benefits of optimizing both trigger value and placement.
- Claim 2: $DPOT_{L_0}$ undermines state-of-the-art defenses. The evidence includes:
    - Extensive Defense Evaluation (Section 6.5 and Appendix D, G, H, I): The paper tests against a wide array of defenses, including robust aggregation methods, outlier detection, and adversarial training.
- Claim 3: $DPOT_{L_0}$ preserves the global model's main-task performance The experiments show that the MA (main-task accuracy) stays within a reasonable range.

**Essential References Not Discussed:**

While the paper mentions adversarial examples (Szegedy, 2014; Carlini & Wagner, 2017), a more in-depth discussion of the connection between $L_0$ attacks and backdoor attacks could be beneficial.

**Experimental Designs Or Analyses:**

- Datasets: Four standard image datasets are used (FashionMNIST, FEMNIST, CIFAR10, Tiny ImageNet), providing a good range of complexity.
- Models: ResNet and VGGNet architectures are used, which are common choices for image classification.
- Defenses: A comprehensive set of defenses is considered, including robust aggregation, and outlier detection.
- Baselines: Relevant baselines are included for comparison.
- Ablation Studies: The paper contains limited studies of MCR and Data poison rate.
- Parameter Settings: The paper provides details on key parameters (MCR, DPR, trigger size, etc.).

**Methods And Evaluation Criteria:**

- Methods:
    - Trigger Optimization (Algorithms 1 & 2): Algorithm 1 finds the trigger locations (pixels with the largest gradient magnitude), and Algorithm 2 optimizes the trigger values using gradient descent. The use of a "trigger training dataset" to enhance generalization is a good design choice.
    - Data Poisoning: The optimized trigger is added to a subset of the malicious clients' data.
- Evaluation Criteria:
    - Attack Success Rate (ASR): Standard metric for evaluating backdoor attacks.
    - Main-task Accuracy (MA): Crucial to show that the attack doesn't completely degrade the model's performance on the main task.
    - Subtlety: Measured by the accuracy drop of a clean model on poisoned data.
    - Comparison to Baselines: The paper compares to several relevant baselines, including fixed-pattern triggers, distributed triggers, and optimized triggers ($L_2$ and $L_0$).

**Other Comments Or Suggestions:**

Consider adding a discussion of potential countermeasures against $DPOT_{L_0}$. Even if the paper focuses on the attack, briefly mentioning possible defense strategies would strengthen the work.

**Other Strengths And Weaknesses:**

- Strengths:
    - The extensive supplementary material and the promise to release code enhance reproducibility.
    - The paper is generally well-written and easy to follow. The algorithms are clearly presented, and the experimental setup is well-described.

- Weaknesses:
    - The theoretical justification relies on a linearity assumption, which limits its direct applicability to neural networks.

**Questions For Authors:**

- While Proposition 5.1 provides a theoretical foundation for linear models, how well do you expect this result to generalize to non-linear neural networks? Have you considered any empirical ways to validate the bound in the non-linear case?
- The paper uses a **trigger training dataset** (D) to optimize the trigger. How sensitive is the attack's performance to the choice of this dataset? Does the trigger generalize well to unseen data from the same distribution? What if the distribution of the trigger training data is different from the distribution of the data seen during FL training?
- How are the attackers selected in each communication round, how many rounds are necessary to achieve the attack goal, and is it required for them to carry out the attack in consecutive rounds?
- How durable is $DPOT_{L_0}$ in comparison to other backdoor attacks (after how many training rounds does the ASR significantly decrease, i.e., by 50% or more)?

**Relation To Broader Scientific Literature:**

- The paper does a good job of citing relevant prior work in backdoor attacks, defenses, and federated learning. Key papers like Bagdasaryan et al. (2020), Sun et al. (2019), and Nguyen et al. (2024) are cited and discussed.
- The paper clearly differentiates itself from existing work by focusing on $L_0$-norm-bounded triggers that optimize both value and placement, and by demonstrating effectiveness through data poisoning alone.

**Theoretical Claims:**

Proposition 5.1 (Concealment Property): This is the main theoretical result. It provides a bound on the difference between the update directions for benign and malicious objectives in a linear model (The proof is in the appendix B.1).

---

> ### Author Rebuttal · Authors · 2025-03-28
>
> Thank you for the effort you put into providing feedback and advice!
>
> 1. Limited studies of MCR and Data poison rate
>
> Due to the space limitation, we discussed ablation studies of MCR  and Data poison rate in Appendix L and N. We also discussed ablation studies of Trigger size and Non-iid degree in Appendix M and O.
>
> 2. Potential countermeasures against $DPOT_{L_0}$
>
> The real issue exposed by crafty data-poisoning attacks like $DPOT_{L_0}$ is a private data governance problem—how to supervise clients' private data to ensure alignment with the common goal of the entire FL system.  Addressing this would need a security framework to protect the entire FL lifespan—from clients deciding to form an FL group to the completion of their FL training. Ensuring both privacy and security may demand system architecture or infrastructure-level interventions beyond ML solutions. We hope to see such frameworks developed collaboratively by the ML and system communities to make FL more practical in real-world applications.
>
> 3. Connection between $L_0$ attacks and backdoor attacks
>
> Can we understand the $L_0$ attacks as $L_0$-norm-bounded adversarial examples used in testing? We cited Papernot et al. (2016) (line 83) for reference. The similarity of $L_0$ adversarial example and $L_0$ backdoor example is that they both constrain the number of altered pixels. The difference is that adversarial examples vary pixel changes per image, while backdoor examples use consistent patterns (shape, value, and placement) to embed a learnable backdoor feature. Unlike adversarial examples, backdoor examples should avoid introducing excessive new features to prevent hindering the main task's convergence.
>
> 4. Empirically validate the bound in the non-linear case
>
> We generated 20 random data points (each has 50 dimensions) to simulate $K$ and 20 random target values $y$ for them. $K_{adv}$ contains 5 random points with specified targets $y_{adv}$ and $\epsilon_{adv} = w(K_{adv}) - y_{adv}$, where $w$ is a model with random parameters. We defined $\Delta w_K = \frac{\partial \Vert w(K) - y\Vert_2}{\partial w}$ and $\Delta w_{K \cup K_{adv}} = \frac{\partial\Vert w([K, K_{adv}]) - [y, y_{adv}]\Vert_2}{\partial w}$, and studied the relationship between $\Vert\epsilon_{adv}\Vert_2$ and their distance $\Vert\Delta w_K - \Delta w_{K \cup K_{adv}}\Vert_2$.
>
> We tested three models: Linear, Two-layer (ReLU activation between linear layers), and LeNet5 (2 convolutional and 3 linear layers). More details are provided [here](https://i.postimg.cc/y8fSVbNc/model-architectures.png).
>
> Varying $\Vert\epsilon_{adv}\Vert_2$ by changing $y_{adv}$, we recorded the corresponding gradient distance changing along with it. The results for different model architectures are presented in this [figure](https://i.postimg.cc/Gm3ynCbb/model-compare-results.png), all of which align with Proposition 5.1. For LeNet5, we also demonstrated the gradient distances separately for its convolutional layers and linear layers in this [figure](https://i.postimg.cc/4NQ90dk8/layer-compare-results.png), finding both bounded by $\Vert\epsilon_{adv}\Vert_2$ with different coefficients. These results indicate that Proposition 5.1 can be applied to non-linear neural networks.
>
> 5. Trigger training data from different distributions
>
> To study this insightful proposal, we constructed Trigger Training Datasets (TTDs) using Out-Of-Distribution (OOD) data and evaluated the impact on $DPOT_{L_0}$'s performance. For the Fashion MNIST task, we used MNIST data to train triggers, and for CIFAR10, we selected data from CIFAR100 that are in different categories to CIFAR10. Triggers generated by OOD data can be found [here](https://i.postimg.cc/CKzndsZt/OOD-ID-visual.png), and the results comparing In-Distribution (ID) and OOD TTDs are shown [here](https://i.postimg.cc/rFk4g4b0/OOD-ID-results.png).
>
> Triggers generated from OOD data might not be the optimal ones to apply to ID victim data, therefore we saw OOD TTDs producing less effective results compared to ID TTDs. Nonetheless, they still show strong attack performance, making this a promising direction for further exploration.
>
> 6. Attack in non-consecutive rounds
>
> In our main evaluation, a few pre-selected attackers participate in every FL round. We also tested non-consecutive attacks, discussed in Appendix P, and found that non-consecutive attacks can enhance $DPOT_{L_0}$'s effectiveness against certain defenses.
>
> 7. The number of rounds needed to achieve the attack goal
>
> We set $ASR$ = 50% as the attack goal and recorded the number of rounds required for different attacks to achieve it under various training tasks and defenses.  The results can be found via this [Link](https://i.postimg.cc/CMjBZ11F/Round-ASR50.png). $DPOT_{L_0}$ shows the fastest speed in reaching the attack goal among all attacks.
>
> 8. Duration of attack effectiveness
>
> Please see our response to Reviewer KkLM, section 3.

---

### Decision · Program_Chairs · 2025-05-01

**Decision:**

Reject

**Comment:**

This paper proposed a new backdoor attack method, DPOT$_{L_0}$, against FL systems that constructs a per-round dynamic, L0-norm objective to bound the generated trigger. Three reviewers voted weak accepts and one voted accept. The authors have responded well to most concerns from the reviewers. The major concerns, which prevent some reviewers from championing the paper, are:

* The evaluation does not reflect real-world setting -- specifically, most experiments are constructed on the ability of the attacker to participate in every round, which is usually not true in practice. While the authors pointed to the experiments in Appendix P, the reviewers also believed that this is not enough. This lack of practical evaluations in several main experiments is a concern.

* The trigger is highly visible, which is a concern. In my opinion, this also raises the question about the effectiveness against defenses such as FLIP, which clearly shows that the method is not effective w.r.t ASR. While accuracy drops under FLIP, this is less than 4%. This raises questions regarding whether DPOT$_{L_0}$'s risks are well understood against these types of defenses. The experiments on 1 dataset, FashionMNIST, do not sufficiently warrant a conclusion.

For these reasons, I believe that the paper is not yet ready for a publication as there are concerns about the real practical risks of the proposed method, even though the averaged rating of this paper is slightly above the borderline.